# Deep Nonlinear Stochastic Optimal Control for Systems with Multiplicative Uncertainties

## Abstract

We present a deep recurrent neural network architecture to solve a class of stochastic optimal control problems described by fully nonlinear Hamilton Jacobi Bellman partial differential equations. Such PDEs arise when one considers stochastic dynamics characterized by uncertainties that are additive, state dependent and control multiplicative. Stochastic models with the aforementioned characteristics have been used in computational neuroscience, biology, finance and aerospace systems and provide a more accurate representation of actuation than models with only additive uncertainty. Previous literature has established the inadequacy of the linear HJB theory and instead relies on a non-linear Feynman-Kac lemma resulting in a second order Forward-Backward Stochastic Differential Equations representation. However, the proposed solutions that use this representation suffer from compounding errors and computational complexity resulting in lack of scalability. In this paper, we propose a deep learning based algorithm that leverages the second order Forward-Backward SDE representation and LSTM based recurrent neural networks to not only solve such Stochastic Optimal Control problems but also overcome the problems faced by traditional approaches and shows promising scalability. The resulting control algorithm is tested on nonlinear systems from robotics and biomechanics in simulation to demonstrate feasibility and out-performance against previous methods.

## 1 Introduction

Stochastic Optimal Control (SOC) is the center of decision making under uncertainty with a long history and extensive prior work both in terms of theory as well as algorithms (Stengel, 1994; Fleming & Soner, 2006). One of the most celebrated formulations of stochastic optimal control is for linear dynamics and additive noise, known as the Linear Quadratic Gaussian (LQG) case (Stengel, 1994). For stochastic systems that are nonlinear in the state and affine in control, the stochastic optimal control formulation results in the Hamilton-Jacobi-Bellman (HJB) equation, which is a backward nonlinear Partial Differential Equation (PDE). Solving the HJB PDE for high dimensional systems is generally a challenging task and suffers from the curse of dimensionality.

Different algorithms have been derived to address stochastic optimal control problems and solve the HJB equation. These algorithms can be mostly classified into two groups: (i) algorithms that rely on linearization and (ii) algorithms that rely on sampling. Linearization-based algorithms rely on first order Taylor's approximation of dynamics (iterative LQG or iLQG) or quadratic approximation of dynamics (Stochastic Differential Dynamic Programming), and quadratic approximation of the cost function (Todorov & Li, 2005b; Theodorou et al., 2010b). Application of the aforementioned algorithms is not straightforward and requires very small time discretization or special linearization schemes especially for the cases of control and/or state dependent noise. It is also worth mentioning that the convergence properties of these algorithms have not been investigated and remain open questions. Sampling-based methods include the Markov-Chain Monte Carlo (MCMC) approximation of the HJB equation (Kushner & Dupuis, 1992; Huynh et al., 2016). MCMC-based algorithms rely on backward propagation of the value function on a pre-specified grid. Recently researchers have incorporated tensor-train decomposition techniques to scale these methods (Gorodetsky et al., 2015). However, these techniques have been applied to special classes of systems and stochastic control

problem formulations and have demonstrated limited applicability so far. Other methods include transforming the SOC problem into a deterministic one by working with the Fokker-Planck PDE representation of the system dynamics (Annunziato & Borzi, 2010; Annunziato & Borzì, 2013). This approach requires solving the problem on a grid as well and therefore doesn't scale to high dimensional systems due to the aforementioned curse of dimensionality.

Alternative sampling-based methodologies rely on the probabilistic representation of the backward PDEs and generalization of the so-called linear Feynman-Kac lemma (Karatzas & Shreve, 1991) to its nonlinear version (Pardoux & Rascanu, 2014). Application of the linear Feynman-Kac lemma requires the exponential transformation of the value function and certain assumptions related to the control cost matrix and the variance of the noise. Controls are then computed using forward sampling of stochastic differential equations (Kappen, 2005; Todorov, 2007; 2009; Theodorou et al., 2010a). The nonlinear version of the Feynman-Kac lemma overcomes the aforementioned limitations. However it requires a more sophisticated numerical scheme than just forward sampling, which relies on the theory of Forward-Backward Stochastic Differential Equations (FBSDEs) and their connection to backward PDEs. The FBSDE formulation is very general and has been utilized in many problem formulations such as $L_2$ and $L_1$ stochastic control (Exarchos & Theodorou, 2018; 2016; Exarchos et al., 2018), min-max and risk-sensitive control (Exarchos et al., 2019) and control of systems with control multiplicative noise (Bakshi et al., 2017). The major limitation of the algorithms that rely on FBSDEs, is the compounding of errors from Least Squares approximation used at every timestep of the Backward Stochastic Differential Equation (BSDE).

Recent efforts in the area of Deep Learning for solving nonlinear PDEs have demonstrated encouraging results in terms of scalability and numerical efficiency. A Deep Learning-based algorithm was introduced by Han et al. (2018) to approximate the solution of non-linear parabolic PDEs through their connection to first order FBSDEs. However, the algorithm was tested on a simple high-dimensional linear system for which an analytical solution already exists. A more recent approach by Pereira et al. (2019) extended this work with a more efficient deep learning architecture and successfully applied the algorithm for control tasks on non-linear systems. However, similar to Han et al. (2016), the approach is only applicable to stochastic systems wherein noise is either additive or state dependent.

In this paper, we develop a novel Deep Neural Network (DNN) architecture for HJB PDEs that correspond to SOC problems in which noise is not only additive or state-dependent but control multiplicative as well. Such problem formulations are important in biomechanics and computational neuroscience, autonomous systems, and finance (Todorov & Li, 2005a; Mitrovic et al., 2010; Primbs, 2007; McLane, 1971; Phillis, 1985). Prior work on stochastic optimal control of such systems considers linear dynamics and quadratic cost functions. Attempts to generalize these linear methods to the case of stochastic nonlinear dynamics with control multiplicative noise are only primitive and require special treatment in terms of methods to forward propagate and linearize the underlying stochastic dynamics (Torre & Theodorou., 2015). Aforementioned probabilistic methods relying on the linear Feynman-Kac lemmma cannot be derived in case of control multiplicative noise.

Below we summarize the contributions of our work:

- We design a novel DNN architecture tailored to solve second-order FBSDEs (2FBSDEs). The neural network architecture consists of Fully Connected (FC) feed-forward and Long-Short Term Memory (LSTM) recurrent layers. The resulting Deep 2FBSDE network can be used to solve fully nonlinear PDEs for nonlinear systems with control multiplicative noise.
- We demonstrate the applicability and correctness of the proposed algorithm in four examples ranging from, traditionally used linear and nonlinear systems in control theory, to robotics and biomechanics. The proposed algorithm recovers analytical controls in the case of linear dynamics while it is also able to successfully control nonlinear dynamics with control-multiplicative and additive sources of uncertainty. Our simulations show the robustness of the Deep 2FBSDE algorithm and prove the importance of considering the nature of the stochastic disturbances in the problem formulation as well as in neural network architecture.

The rest of the paper is organized as follows: in Section 2 we first introduce notation, some preliminaries and discuss the problem formulation. Next, in Section 3 we provide the 2FBSDE formulation. The Deep 2FBSDE algorithm is introduced in Section 4. Then we demonstrate and discuss results from our simulation experiments in Section 5. Finally we conclude the paper in Section 6 with discussion and future directions.

## 2 STOCHASTIC OPTIMAL CONTROL

In this section we provide notations and stochastic optimal control concepts essential for the development of our proposed algorithm and then present the problem formulation. Note that the bold faced notation will be abused for both vectors and matrices, while non-bold faced will be used for scalars.

### 2.1 PRELIMINARIES

We first introduce stochastic dynamical systems which have a *drift* component (i.e. the non-stochastic component of the dynamics) that is a nonlinear function of the state but affine with respect to the controls. The stochastic component is comprised of nonlinear functions of the state and affine control multiplicative matrix coefficients. Let $\boldsymbol{x} \in \mathbb{R}^{n_x}$ be the vector of state variables, and $\boldsymbol{u} \in \mathbb{R}^{n_u}$ be the vector of control variables taking values in the set of all admissible controls $\mathcal{U}([0,T])$, for a fixed finite time horizon $T \in [0, \infty)$. Let $\left([v(t)^{\mathrm{T}} \; \boldsymbol{w}(t)^{\mathrm{T}}]^{\mathrm{T}}\right)_{t \in [0,T]}$ be a Brownian motion in $\mathbb{R}^{n_w+1}$, where $v(t) \in \mathbb{R}$, $\boldsymbol{w}(t) \in \mathbb{R}^{n_w}$ and the components of $\boldsymbol{w}(t)$ are mutually independent one dimensional standard Brownian motions. We now assume that functions $\boldsymbol{f} : [0,T] \times \mathbb{R}^{n_x} \to \mathbb{R}^{n_x}$, $\mathbf{G} : [0,T] \times \mathbb{R}^{n_x} \to \mathbb{R}^{n_x \times n_u}$, $\boldsymbol{\Sigma} : [0,T] \times \mathbb{R}^{n_x} \to \mathbb{R}^{n_x \times n_w}$ and $\sigma \in \mathbb{R}^+$ satisfy certain Lipschitz and growth conditions (see supplementary materials (SM) A).

Given this assumption, it is known that for every initial condition $\xi \in \mathbb{R}^{n_x}$, there exists a unique solution $\left(\boldsymbol{x}(t)\right)_{t \in [0,T]}$ to the Forward Stochastic Differential Equation (FSDE),

$$\begin{cases} \mathrm{d}\boldsymbol{x}(t) &= \underbrace{\big(\boldsymbol{f}(t, \boldsymbol{x}(t)) + \mathbf{G}(t, \boldsymbol{x}(t))\boldsymbol{u}(t)\big)}_{\text{drift vector}} \mathrm{d}t + \underbrace{\sigma\mathbf{G}(t, \boldsymbol{x}(t))\boldsymbol{u}(t)\mathrm{d}v(t)}_{\text{control multiplicative noise}} + \underbrace{\boldsymbol{\Sigma}(t, \boldsymbol{x}(t))\mathrm{d}\boldsymbol{w}(t)}_{\text{only state-dependent noise}} \\ &= \big(\boldsymbol{f}(t, \boldsymbol{x}(t)) + \mathbf{G}(t, \boldsymbol{x}(t))\boldsymbol{u}(t)\big)\mathrm{d}t + \mathbf{C}\big(t, \boldsymbol{x}(t), \boldsymbol{u}(t)\big)\mathrm{d}\boldsymbol{\epsilon}(t) \\ \boldsymbol{x}(0) &= \xi, \end{cases} \quad (1)$$

where $\mathbf{C}(t, \boldsymbol{x}(t), \boldsymbol{u}(t)) = \big[\sigma\mathbf{G}(t, \boldsymbol{x}(t))\boldsymbol{u}(t), \; \boldsymbol{\Sigma}(t, \boldsymbol{x}(t))\big]$ and $\boldsymbol{\epsilon}(t) = \begin{bmatrix} v(t) \\ \boldsymbol{w}(t) \end{bmatrix}$.

### 2.2 PROBLEM STATEMENT AND HJB PDE

For the controlled stochastic dynamical system above, we formulate the SOC problem as minimizing the following expected cost quadratic in control

$$J\big(t, \boldsymbol{x}(t); \boldsymbol{u}(t)\big) = \mathbb{E}_{\mathbb{Q}}\left[\int_t^T \big(q(s, \boldsymbol{x}(s)) + \frac{1}{2}\boldsymbol{u}(s)^{\mathrm{T}}\mathbf{R}\boldsymbol{u}(s)\big)\mathrm{d}s + \phi\big(\boldsymbol{x}(T)\big)\right] \quad (2)$$

where $q : \mathbb{R}^{n_x} \to \mathbb{R}^+$ is the running state-dependent cost function, $\mathbf{R}$ (control cost coefficients) is a symmetric positive definite matrix of size $n_u \times n_u$ and $\phi : \mathbb{R}^{n_x} \to \mathbb{R}^+$ is the terminal state cost. $q$ and $\phi$ are differentiable with continuous derivatives up to the second order. The expectation is taken with respect to the probability measure $\mathbb{Q}$ over the space of trajectories induced by the controlled stochastic dynamics in equation 1. We can define the value function as

$$\begin{cases} V\big(t, \boldsymbol{x}(t)\big) &= \inf_{\boldsymbol{u}(t) \in \mathcal{U}([t,T])} J\big(t, \boldsymbol{x}(t); \boldsymbol{u}(t)\big) \\ V\big(T, \boldsymbol{x}(T)\big) &= \phi\big(\boldsymbol{x}(T)\big). \end{cases} \quad (3)$$

Under the condition that the value function is once differentialble in $t$ and twice differentiable in $\boldsymbol{x}$ with continuous derivatives, we can follow the standard stochastic optimal control derivation to obtain the HJB PDE (written without all the functional dependencies for simplicity)

$$\begin{cases} V_t + q + V_{\boldsymbol{x}}^{\mathrm{T}}\boldsymbol{f} - \frac{1}{2}V_{\boldsymbol{x}}^{\mathrm{T}}\mathbf{G}\hat{\mathbf{R}}^{-1}\mathbf{G}^{\mathrm{T}}V_{\boldsymbol{x}} + \frac{1}{2}\,\mathrm{tr}(V_{\boldsymbol{x}\boldsymbol{x}}\boldsymbol{\Sigma}\boldsymbol{\Sigma}^{\mathrm{T}}) = 0 \\ V(T, \boldsymbol{x}(T)) = \phi(\boldsymbol{x}(T)), \end{cases} \quad (4)$$

with the optimal control of the form,

$$\boldsymbol{u}^*(t, \boldsymbol{x}) = -\hat{\mathbf{R}}^{-1}\mathbf{G}(t, \boldsymbol{x})^{\mathrm{T}}V_{\boldsymbol{x}}(t, \boldsymbol{x}). \quad (5)$$

where, $\hat{\mathbf{R}} \triangleq (\mathbf{R} + \sigma^2\mathbf{G}^{\mathrm{T}}V_{\boldsymbol{x}\boldsymbol{x}}\mathbf{G})$. The derivation for both equations 4 and 5 can be found in SM B.

## 3 A FBSDE Solution to the HJB PDE

The theory of BSDEs establishes a connection between the solution of a parabolic PDE and a set of FBSDEs (Pardoux & Peng, 1990; El Karoui et al., 1997). This connection has been used to solve the HJB PDE in the context of SOC problems. Exarchos & Theodorou (2018) solved the HJB PDE in the absence of control multiplicative noise $\mathrm{d}v$ with a set of first order FBSDEs. Bakshi et al. (2017) utilized the second order FBSDEs or 2FBSDEs to solve the fully nonlinear HJB PDE in the presence of control multiplicative noise, but did not consider any control in the FSDE. Lack of control leads to insufficient exploration and for highly nonlinear systems renders it impossible to find an optimal solution to complete the task. In light of this, we introduce a new set of 2FBSDEs

$$
\begin{cases}
\mathrm{d}\boldsymbol{x} & = \boldsymbol{f}\mathrm{d}t + \mathbf{G}\boldsymbol{u}^*\mathrm{d}t + \mathbf{C}\mathrm{d}\boldsymbol{\epsilon} & \text{(FSDE)} \\
\mathrm{d}V & = \mathcal{H}(V)\mathrm{d}t + V_{\boldsymbol{x}}^{\mathrm{T}}\mathbf{G}\boldsymbol{u}^*\mathrm{d}t + V_{\boldsymbol{x}}^{\mathrm{T}}\mathbf{C}\mathrm{d}\boldsymbol{\epsilon} & \text{(BSDE 1)} \\
\mathrm{d}V_{\boldsymbol{x}} & = \mathcal{H}(V_{\boldsymbol{x}})\mathrm{d}t + V_{\boldsymbol{x}\boldsymbol{x}}\mathbf{G}\boldsymbol{u}^*\mathrm{d}t + V_{\boldsymbol{x}\boldsymbol{x}}\mathbf{C}\mathrm{d}\boldsymbol{\epsilon} & \text{(BSDE 2)} \\
\boldsymbol{x}(0) & = \xi,\ V(T) = \phi(\boldsymbol{x}(T)),\ V_{\boldsymbol{x}}(T) = \phi_{\boldsymbol{x}}(\boldsymbol{x}(T)),
\end{cases}
\tag{6}
$$

where $\boldsymbol{u}^*$ is the optimal control given by equation 5 and the operator $\mathcal{H}$ is defined as

$$
\mathcal{H}(\cdot) \triangleq \partial_t(\cdot) + \partial_{\boldsymbol{x}}(\cdot)^{\mathrm{T}}\boldsymbol{f} + \frac{1}{2}\mathrm{tr}(\partial_{\boldsymbol{x}\boldsymbol{x}}(\cdot)\mathbf{C}\mathbf{C}^{\mathrm{T}}).
\tag{7}
$$

Note that the functional dependencies are dropped in this and the following section for simplicity. We can obtain this particular set of 2FBSDEs by substituting for the optimal control (equation 5) in the forward process (equation 1). The backward processes are obtained by applying Ito's lemma (Itô, 1944), which is essentially the second order Taylor series expansion, to the value function $V$ and its gradient $V_{\boldsymbol{x}}$. Additionally, we substitute the expression for $V_t$ from equation 4 into BSDE 1 and obtain the final form of the 2FBSDEs (detailed derivation is included in SM C) as

$$
\begin{cases}
\mathrm{d}\boldsymbol{x} & = \boldsymbol{f}\mathrm{d}t + \mathbf{G}(\boldsymbol{u}^*\mathrm{d}t + \sigma\boldsymbol{u}^*\mathrm{d}v) + \boldsymbol{\Sigma}\mathrm{d}\boldsymbol{w} \\
\mathrm{d}V & = -\big(q - \frac{1}{2}V_{\boldsymbol{x}}^{\mathrm{T}}\mathbf{G}\hat{\mathbf{R}}^{-T}(\mathbf{R} + 2\sigma^2\mathbf{G}^{\mathrm{T}}V_{\boldsymbol{x}\boldsymbol{x}}\mathbf{G})\hat{\mathbf{R}}^{-1}\mathbf{G}^{\mathrm{T}}V_{\boldsymbol{x}}\big)\mathrm{d}t \\
& \quad + V_{\boldsymbol{x}}^{\mathrm{T}}\mathbf{G}(\boldsymbol{u}^*\mathrm{d}t + \sigma\boldsymbol{u}^*\mathrm{d}v) + V_{\boldsymbol{x}}^{\mathrm{T}}\boldsymbol{\Sigma}\mathrm{d}\boldsymbol{w} \\
\mathrm{d}V_{\boldsymbol{x}} & = (\mathbf{A} + V_{\boldsymbol{x}\boldsymbol{x}}\boldsymbol{f})\mathrm{d}t + V_{\boldsymbol{x}\boldsymbol{x}}\mathbf{G}(\boldsymbol{u}^*\mathrm{d}t + \sigma\boldsymbol{u}^*\mathrm{d}v) + V_{\boldsymbol{x}\boldsymbol{x}}\boldsymbol{\Sigma}\mathrm{d}\boldsymbol{w} \\
\boldsymbol{x}(0) & = \xi,\ V(T) = \phi(\boldsymbol{x}(T)),\ V_{\boldsymbol{x}}(T) = \phi_{\boldsymbol{x}}(\boldsymbol{x}(T)),
\end{cases}
\tag{8}
$$

with $\mathbf{A} = \partial_t(V_{\boldsymbol{x}}) + \frac{1}{2}\mathrm{tr}(\partial_{\boldsymbol{x}\boldsymbol{x}}(V_{\boldsymbol{x}})\mathbf{C}\mathbf{C}^{\mathrm{T}})$.

Using equation 8, we can forward sample the FSDE. The second-order BSDEs (2BSDEs), on the other hand, need to satisfy a terminal condition and therefore have to be propagated backwards in time. However, since the uncertainty that enters the dynamics evolves forward in time, only the conditional expectations of the 2BSDEs can be back-propagated. For more details please refer to Shreve (2004, Chapter 2). Bakshi et al. (2017) back-propagate approximate conditional expectations, computed using regression, of the two processes. This method however, suffers from compounding errors introduced by least squares estimation at every time step. In contrast a Deep Learning (DL) based approach, first introduced by Han et al. (2018), mitigates this problem by using the terminal condition as the prediction target for a forward propagated BSDE. This is enabled by randomly initializing the value function and its gradient at the start time and treating them as trainable parameters of a self-supervised deep learning problem. In addition, the approximation errors at each time step are compensated for by backpropagation during training of the DNN. This allowed using FBSDEs to solve the HJB PDE for high-dimensional linear systems. A more recent approach, the Deep FBSDE controller (Pereira et al., 2019), futher extends the work successfully to nonlinear systems, with and without control constraints, in simulation that correspond to first order FBSDEs. It also introduced a LSTM-based network architecture, in contrast to using separate feed-forward neural networks at every timestep as in Han et al. (2016), that resulted in superior performance, memory and time complexity. Extending this work, we propose a new framework for solving SOC problems of systems with control multiplicative noise, for which the value function solutions correspond to 2FBSDEs.

## 4 Deep 2FBSDE Controller

In this section, we introduce a new deep network architecture called the *Deep 2FBSDE Controller* and present a training algorithm to solve SOC problems with control multiplicative noise.

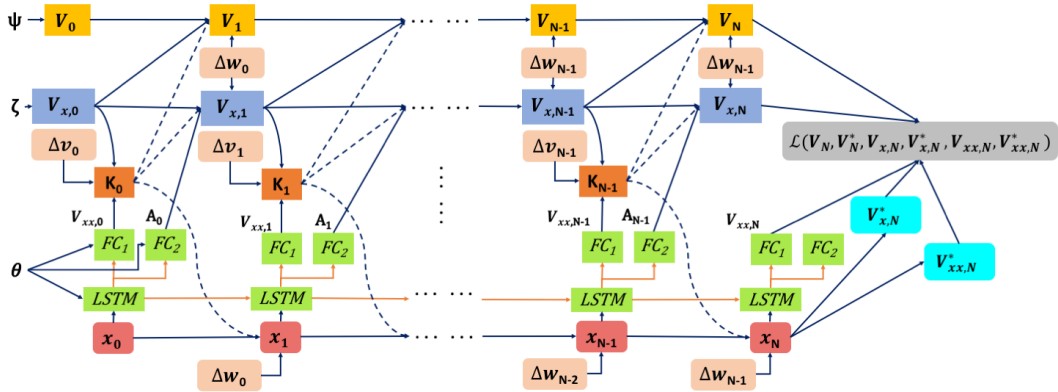

Figure 1: **Deep 2FBSDE neural network architecture.** The blocks $FC_{1,2}$ are fully connected layers with linear activations while the $LSTM$ block represents recurrent layers of stacked LSTM cells with standard nonlinear activations. These layers are parameterized by $\theta$ and shared temporally. Additionally, $V(\boldsymbol{x}_0, t_0)$ and $V_{\boldsymbol{x}}(\boldsymbol{x}, t_0)$ are parameterized by $\psi$ and $\zeta$ respectively. The self-supervised targets $V_N^*, V_{\boldsymbol{x},N}^*$ are the terminal conditions from equation 8 and $V_{\boldsymbol{xx},N}^* = \phi_{\boldsymbol{xx}}(\boldsymbol{x}_N)$.

**Time discretization:** In order to approximate numerical solutions of the Stochastic Differential Equations (SDEs) we choose the explicit Euler-Maruyama time-discretization scheme (Kloeden & Platen, 2013, Chapter 9) similar to (Pereira et al., 2019; Han et al., 2018). Here we overload $t$ as both the continuous-time variable and discrete time index and discretize the task horizon $0 < t < T$ as $t = \{0, 1, \cdots, N\}$, where $T = N\Delta t$. This is also used to discretize all variables as step functions if their discrete time index $t$ lies in the interval $[t\Delta t, (t+1)\Delta t]$. We use subscript $t$ to denote the discretized variables in our proposed algorithm Alg. 1.

---

**Algorithm 1:** Finite Time Horizon Deep 2FBSDE Controller

**Given:** $\xi, \boldsymbol{f}, \mathbf{G}, \boldsymbol{\Sigma}, \sigma$: Initial state, drift, actuation dynamics, state dependent noise matrix and control multiplicative noise std. deviation; $\phi, q, \mathbf{R}$: Cost function parameters; $N$: Task horizon, $K$: Number of iterations, $M$: Batch size; $\Delta t$: Time discretization; $\lambda$: regularization parameter;
**Parameters:** $V_0 = V(\boldsymbol{x}_0; \psi)$: Value function at $t = 0$ parameterized by trainable weight $\psi$;
$V_{\boldsymbol{x},0} = V_{\boldsymbol{x}}(\boldsymbol{x}; \zeta)$: Gradient of value function at $t = 0$ parameterized by trainable weights $\zeta$;
$\theta$: Weights and biases of all fully-connected and LSTM layers;
**Initialize all trainable paramters and states:** $\theta^0, \psi^0, \zeta^0, \boldsymbol{x}_0 = \xi$
**for** $k = 1$ **to** $K$ **do**
  **for** $i = 1$ **to** $M$ **do**
    **for** $t = 0$ **to** $N - 1$ **do**
      Compute $\mathbf{G}$ matrix: $\mathbf{G}_t^i = \mathbf{G}(\boldsymbol{x}_t^i, t)$;
      Network prediction: $V_{\boldsymbol{xx},t}^i, \mathbf{A}_t^i = f_{FC_1}(f_{LSTM}(\boldsymbol{x}; \theta^{k-1})), f_{FC_2}(f_{LSTM}(\boldsymbol{x}; \theta^{k-1}))$
      Compute optimal control: $\boldsymbol{u}_t^{i*} = -(\mathbf{R} + \sigma^2 \mathbf{G}_t^{i\mathrm{T}} V_{\boldsymbol{xx},t}^i \mathbf{G}_t^i)^{-1} \mathbf{G}_t^{i\mathrm{T}} V_{\boldsymbol{x},t}^i$;
      Sample Brownian noise: $\Delta \boldsymbol{w}_t^i \sim \mathcal{N}(0, \mathbf{I}\Delta t); \Delta v_t^i \sim \mathcal{N}(0, \Delta t)$
      Compute control: $\mathbf{K}_t^i = \mathbf{K}(t, \boldsymbol{x}_t^i) = \mathbf{G}_t^i(\boldsymbol{u}_t^{i*}\Delta t + \sigma \boldsymbol{u}_t^{i*}\Delta v_t^i)$
      Forward propagate the SDEs:
      $\boldsymbol{x}_{t+1}^i, V_{t+1}^i, V_{\boldsymbol{x},t+1}^i = f_{2FBSDE}(\boldsymbol{x}_t^i, V_t^i, V_{\boldsymbol{x},t}^i, V_{\boldsymbol{xx},t}^i, \mathbf{K}_t^i) \leftarrow$ time discretized equation 8
    **end for**
  **end for**
  Compute mini-batch loss: $\mathcal{L} = f_{Loss}(V_N, V_N^*, V_{\boldsymbol{x},N}, V_{\boldsymbol{x},N}^*, V_{\boldsymbol{xx},N}, V_{\boldsymbol{xx},N}^*, \theta^{k-1})$
  Gradient update: $\theta^k, \psi^k, \zeta^k \leftarrow \text{Adam.step}(\mathcal{L}, \theta^{k-1}, \psi^{k-1}, \zeta^{k-1})$
**end for**
**return** $\theta^K, \psi^K, \zeta^K$

---

**Network architecture:** Inspired by the LSTM-based recurrent neural network architecture introduced by Pereira et al. (2019) for solving first order FBSDEs, we propose the network in fig.1 adapted to the uncertainties in 2FBSDEs given by equation 8. Instead of predicting the gradient of the value function $V_{\boldsymbol{x}}$ at every time step, the output of the LSTM is used to predict the Hessian of the value function $V_{\boldsymbol{xx}}$ and $\mathbf{A} = \partial_t(V_{\boldsymbol{x}}) + \frac{1}{2}\text{tr}(\partial_{\boldsymbol{xx}}(V_{\boldsymbol{x}})\mathbf{C}\mathbf{C}^{\mathrm{T}})$ using two separate FC layers with

linear activations. Notice that $\partial_{\boldsymbol{xx}}(V_{\boldsymbol{x}})$ is a rank 3 tensor and using neural networks to predict this term explicitly would render this method unscalable. We, however, bypass this problem by instead predicting the trace of the tensor product which is a vector allowing linear growth in output size with state dimensionality. Of these, $V_{\boldsymbol{xx}}$ is used to compute the control term $\mathbf{K} = \mathbf{G}(\boldsymbol{u}^*\mathrm{d}t + \sigma\boldsymbol{u}^*\mathrm{d}v)$. This in turn is used to propagate the stochastic dynamics in equation 8. Both $V_{\boldsymbol{xx}}$ and $\mathbf{A}$ are used to propagate $V_{\boldsymbol{x}}$ which is then used to propagate $V$. This is repeated until the end of the time horizon as shown in fig.1. Finally, the predicted values of $V$, $V_{\boldsymbol{x}}$ and $V_{\boldsymbol{xx}}$ are compared with their targets computed using $\boldsymbol{x}$ at the end of the horizon, to compute a loss function for backpropagation.

**Algorithm:** Algorithm 1 details the training procedure of the Deep 2FBSDE Controller. Note that superscripts indicate batch index for variables and iteration number for trainable parameters. The value function $V_0$ and its gradient $V_{\boldsymbol{x},0}$ (at time index $t = 0$), are randomly initialized and trainable. Functions $f_{LSTM}, f_{FC_1}$ and $f_{FC_2}$ denote the forward propagation equations of standard LSTM layers (Hochreiter & Schmidhuber, 1997) and FC layers with tanh and linear activations respectively, and $f_{2FBSDE}$ represents a discretized version of equation 8 using the explicit Euler-Maruyama time discretization scheme. The loss function ($\mathcal{L}$) is computed using the given terminal conditions as targets, the propagated value function ($V$), its gradient ($V_{\boldsymbol{x}}$) and Hessian ($V_{\boldsymbol{xx}}$) at the final time index as well as an $L_2$ regularization term as follows

$$\mathcal{L} = c_1\,||V^* - V_N||_2^2 + c_2\,||V_{\boldsymbol{x}}^* - V_{\boldsymbol{x},N}||_2^2 + c_3\,||V_{\boldsymbol{xx}}^* - V_{\boldsymbol{xx},N}||_2^2 + c_4\,||V^*||_2^2 + \lambda||\theta||_2^2. \quad (9)$$

A detailed justification of each loss term is included in SM D. The network is trained using any variant of Stochastic Gradient Descent (SGD) such as Adam (Kingma & Ba, 2014) until convergence. The algorithm returns a trained network capable of computing an optimal feedback control at every timestep starting from the given initial state.

## 5 SIMULATION RESULTS

In this section we demonstrate the capability of the Deep 2FBSDE Controller on 4 different systems in simulation. We first compare the solution of the Deep 2FBSDE Controller to the analytical solution for a scalar linear system to validate the correctness of our proposed algorithm. We then consider a cart-pole swing-up task and a reaching task for a 12-state quadcopter. Finally, we tested on a 2-link 6-muscle (10-state) human arm model for a planar reaching task. The results were compared against the Deep FBSDE Controller in Pereira et al. (2019), where in the effect of control multiplicative noise was ignored by only considering additive noise models resulting in first order FBSDEs, and the iLQG controller in Li & Todorov (2007). Hereon we use 2FBSDE, FBSDE and iLQG to denote the Deep 2FBSDE, Deep FBSDE and iLQG controllers respectively. The system dynamics can be found in SM E and the simulation parameters can be found in SM F.

All the comparison plots contain statistics gathered over 128 test trials. For 2FBSDE and FBSDE, we used time discretization $\Delta t = 0.004$ seconds for the linear system problem, $\Delta t = 0.02$ seconds for the cart-pole and quadcopter simulations and $\Delta t = 0.02$ seconds for the human arm simulation. For the iLQG simulations, $\Delta t = 0.01$ seconds for cartpole and quadcopter and $\Delta t = 0.001$ seconds for the human arm were used to avoid numerical instability. These values were hand-tuned until reasonable performance was observed from iLQG. In all plots, the solid lines represent mean trajectories, and the shaded regions represent the 68% confidence region (mean $\pm$ standard deviation).

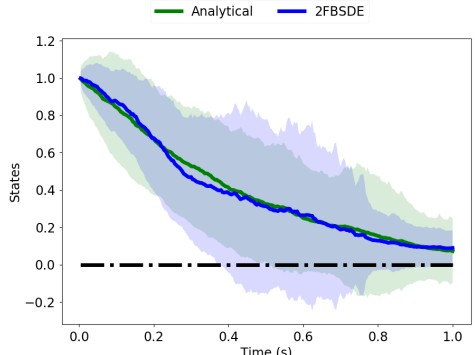

Figure 2: The 2FBSDE solution is very similar to the analytical solution for a scalar linear system.

**Linear System:** We consider a scalar linear time-invariant system

$$\mathrm{d}x(t) = Ax(t)\mathrm{d}t + Bu(t)\mathrm{d}t + \sigma Bu(t)\mathrm{d}v(t) + C\mathrm{d}w(t), \quad (10)$$

with quadratic running state cost and terminal state cost $q\big(x(t)\big) = \frac{1}{2}Qx(t)^2, \phi\big(x(T)\big) = \frac{1}{2}Q_Tx(t)^2$.

The dynamics and cost function parameters are set as $A = 0.2, B = 1.0, C = 0.1, \sigma = 0.5, Q = 0, Q_T = 80, R = 2, x_0 = 1.0$. The task is to drive the state to 0. Note that the problem is different

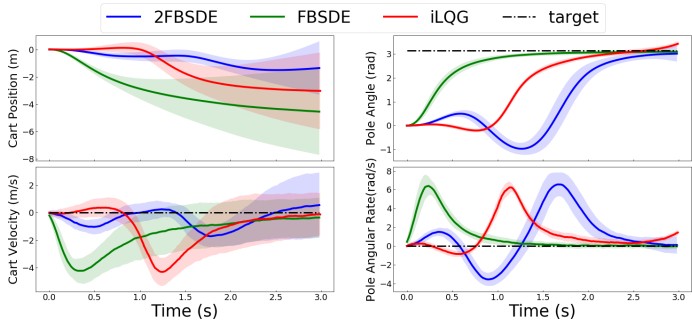

Figure 3: **Cartpole Swing-up Task:** 2FBSDE is most aware of the presence of control multiplicative noise and uses least control effort to perform the task. This is evident from the cart velocity plot, a directly actuated state, which tries to stay as close to zero as possible.

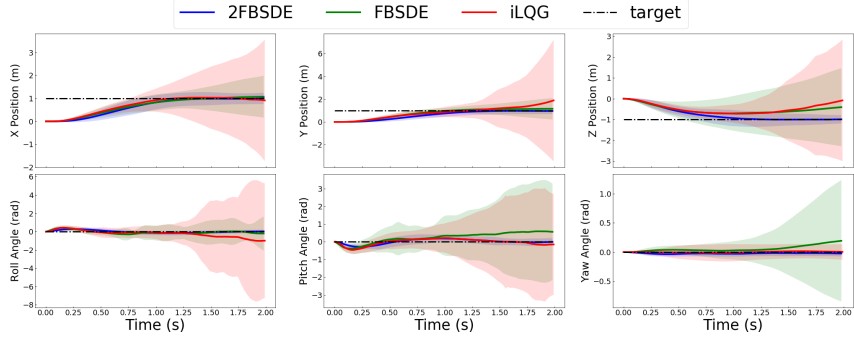

Figure 4: **Quadcopter Reach and Hover task:** 2FBSDE clearly out-performs both FBSDE and iLQG, as indicated by the smaller variance in all states.

than the one for LQG due to the presence of control multiplicative noise (i.e. $v(t) \neq 0$). Let us assume that the value function (equation 4) has the form $V\big(t, x(t)\big) = \frac{1}{2}P(t)x^2(t) + S(t)x(t) + c(t)$, where $S(T) = 0, P(T) = Q_T, c(t) = \int_0^T \frac{1}{2}c^2 P(s)\mathrm{d}s$. The values of $P$ and $Q$ are obtained by solving corresponding Riccatti equations, using ODE solvers, that can be derived in a similar manner to the LQG case in Stengel (1994, Chapter 5) and are used to compute the optimal control (equation 5) at every timestep. The solution obtained from 2FBSDE is compared against the analytical solution in fig. 2. The resulting trajectories have matching mean and comparable variance, which verifies the effectiveness of the controller on linear systems.

**Cartpole:** We applied the controllers to cartpole dynamics for a swing-up task with a time horizon of $T = 2.0$ seconds. The networks were trained using a batch size of 256 each for 2000 iterations. We imposed a strict restriction on the controllers with control cost coefficient $R = 0.5$ and did not impose any cost or target for the cart position state. We would like to highlight in fig. 3 that both 2FBSDE and iLQG choose to pre-swing the pendulum and use the system's momentum to achieve the swing-up task. They thus respect the presence of control multiplicative noise entering the system as compared to FBSDE which tries to drive the pendulum to the desired vertical position as soon as possible by applying as much control as required. Additionally, we tested for a lower control cost $R = 0.1$ (see SM G for plots) and observed qualitatively different behavior from the controllers.

**Quadcopter:** The controller was also tested on a 12-state quadcopter dynamics model for a task of reaching and hovering at a target position with a time horizon of 2.0 seconds. The network was trained with a batch size of 256 for 5000 iterations. Only linear and angular states are included in fig. 4 since they most directly reflect the task performance (velocity plots included in SM G). The figure demonstrates superior performance of the 2FBSDE controller over the FBSDE and iLQG controller in reaching the target state faster and maintaining smaller state variance. Moreover, these results also convey the importance of taking into account multiplicative noise in the design of optimal controllers as the state dimensionality and system complexity increases. We also tested the 2FBSDE controller

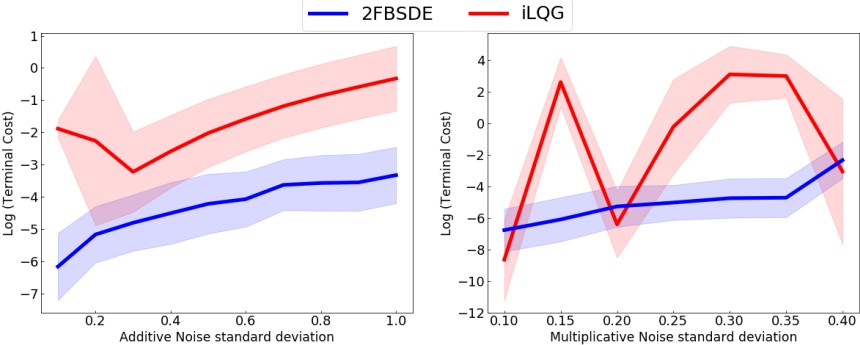

Figure 5: **Comparison of 2FBSDE and iLQG for Human Arm system:** Plots show terminal cost versus additive ($\Sigma$) and control multiplicative noise standard deviations ($\sigma$). The FBSDE results are omitted as failure (divergence) occurred in all test trials. When additive noise is varied (*left*), 2FBSDE outperforms iLQG at lower $\Sigma$ and performance deteriorates much slower than iLQG as $\Sigma$ increases. When multiplicative noise is varied (*right*), iLQG exhibits very erratic behaviour while the terminal cost gradually increases with $\sigma$ for 2FBSDE.

for the same task (reach and hover) for a longer horizon of 3.0 seconds (see SM G for plots) to verify its stabilizing capability in presence of control multiplicative noise.

**2-link 6-muscle Human Arm:** This is a 10-state bio-mechanical system wherein control multiplicative noise models have been found to closely mimic empirical observations (Todorov, 2005). The controllers were tasked with reaching a target position in 1.0 second. The networks were trained with a batch size of 256 for 2000 iterations. Additive noise in the joint-angle acceleration channels and multiplicative noise in muscle activations were considered. In fig. 5, we show the log of terminal state cost ($\log \phi(\boldsymbol{x}_N)$) versus different values of additive noise standard deviations (while holding multiplicative noise standard deviation fixed at 0.1) on the left and versus different values of multiplicative noise standard deviations (keeping additive noise standard deviation fixed at 0) on the right. As seen in the fig. 5, the performance of iLQG is very erratic compared to 2FBSDE as $\sigma$ is varied. The difference in behaviors can be attributed to the fact that iLQG is only aware of the first two moments of the uncertainty entering the system while the 2FBSDE, being a sampling based controller is exposed to the true uncertainty entering the system.

**Discussion:** Although the performance of iLQG for the cartpole and human arm tasks seem competitive to 2FBSDE, we would like to highlight the fact that iLQG becomes brittle as $\sigma$ of the control multiplicative noise is increased (or time horizon is increased) and requires very fine time discretizations, proper regularization scheduling and fine state cost coefficient tuning in order to be able to converge. Moreover, the control cost ($\mathbf{R}$) had to be tuned to a high value in order to prevent divergence at higher noise standard deviations. This is in contrast to the results in Li & Todorov (2007), on account of two main reasons: firstly they do not consider the presence of additive noise in the angular acceleration channels in addition to multiplicative noise in the muscle activation channels and secondly the paper does not provide all details to fully reproduce the published results nor is code with muscle actuation made publicly available. Some of the crucial elements to fully implement the human arm model such as relationships between muscle lengths, muscle velocities, and the respective joint angles had to be obtained by studying the work done by Teka et al. (2017).

## 6 CONCLUSIONS

In this paper, we proposed the Deep 2FBSDE Controller to solve the Stochastic Optimal Control problems for systems with additive, state dependent and control multiplicative noise. The algorithm relies on the 2FBSDE formulation with control in the forward process for sufficient exploration. The effectiveness of the algorithm is demonstrated by comparing against analytical solution for a linear system, and against the first order FBSDE controller and the iLQG controller on systems of cartpole, quadcopter and 2-link human arm in simulation. Potential future directions of this work include application to financial models with intrinsic control multiplicative noise and theoretical analysis of error bounds on value function approximation as well as investigating newer architectures.

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

# SUPPLEMENTARY MATERIALS

## A    ASSUMPTIONS

**Assumption 1.** *There exists a constant $k > 0$ such that*

$$|(\boldsymbol{f} + \mathbf{G}\boldsymbol{u})(t, \boldsymbol{x}, \boldsymbol{u}) - (\boldsymbol{f} + \mathbf{G}\boldsymbol{u})(t, \boldsymbol{x}', \boldsymbol{u}')| \leq k(|\boldsymbol{x} - \boldsymbol{x}'| + |\boldsymbol{u} - \boldsymbol{u}'|) \tag{11}$$

$$|\mathbf{C}(t, \boldsymbol{x}, \boldsymbol{u}) - \mathbf{C}(t, \boldsymbol{x}', \boldsymbol{u}')| \leq k(|\boldsymbol{x} - \boldsymbol{x}'| + |\boldsymbol{u} - \boldsymbol{u}'|) \tag{12}$$

$$|(\boldsymbol{f} + \mathbf{G}\boldsymbol{u})(t, \boldsymbol{x}, \boldsymbol{u}) + \mathbf{C}(t, \boldsymbol{x})| \leq k(1 + |\boldsymbol{x}|), \tag{13}$$

*where* $\mathbf{C} = [\sigma \mathbf{G}\boldsymbol{u} \ \ \boldsymbol{\Sigma}]$, $\forall t \in [0, T]$, $\forall \boldsymbol{x}, \boldsymbol{x}' \in \mathbb{R}^{n_x}$ *and* $\forall \boldsymbol{u}, \boldsymbol{u}' \in \mathbb{R}^{n_u}$.

## B    HJB PDE DERIVATION

Applying the dynamic programming principle (Bellman & Kalaba, 1964) to the value function we have

$$V(t, \boldsymbol{x}(t)) = \inf_{\boldsymbol{u}(s) \in \mathcal{U}([t, t+\mathrm{d}t])} \mathbb{E}_{\mathbb{Q}}\left[ \int_t^{t+\mathrm{d}t} \left(q(\boldsymbol{x}(s)) + \frac{1}{2}\boldsymbol{u}(s)^{\mathrm{T}}\mathbf{R}\boldsymbol{u}(s)\right)\mathrm{d}s + V(t + \mathrm{d}t, \boldsymbol{x}(t + \mathrm{d}t)) \right]. \tag{14}$$

Then, we can approximate the running cost integral with a step function and apply Ito's lemma (Itô, 1944) to obtain

$$
\begin{aligned}
V(t, \boldsymbol{x}(t)) &= \inf_{\boldsymbol{u} \in \mathcal{U}} \mathbb{E}_{\mathbb{Q}}\left[ \left(q(\boldsymbol{x}(t)) + \frac{1}{2}\boldsymbol{u}(t)^{\mathrm{T}}\mathbf{R}\boldsymbol{u}(t)\right)\mathrm{d}t + V(t + \mathrm{d}t, \boldsymbol{x}(t + \mathrm{d}t)) \right] \\
&\stackrel{\text{Itô}}{=} \inf_{\boldsymbol{u} \in \mathcal{U}} \mathbb{E}_{\mathbb{Q}}\left[ \left(q(\boldsymbol{x}(t)) + \frac{1}{2}\boldsymbol{u}(t)^{\mathrm{T}}\mathbf{R}\boldsymbol{u}(t)\right)\mathrm{d}t + V(t, \boldsymbol{x}(t)) + V_t(t, \boldsymbol{x}(t))\mathrm{d}t \right. \\
&\quad + V_{\boldsymbol{x}}^{\mathrm{T}}(t, \boldsymbol{x}(t))\Big(\boldsymbol{f}(t, \boldsymbol{x}(t))\mathrm{d}t + \mathbf{G}(t, \boldsymbol{x}(t))(\boldsymbol{u}\mathrm{d}t + \sigma\boldsymbol{u}(t)\mathrm{d}v(t)) \\
&\quad \left. + \boldsymbol{\Sigma}(t, \boldsymbol{x}(t))\mathrm{d}\boldsymbol{w}(t)\Big) + \frac{1}{2}\mathrm{tr}\big(V_{\boldsymbol{x}\boldsymbol{x}}(t, \boldsymbol{x}(t))\mathbf{C}(t, \boldsymbol{x}(t), \boldsymbol{u}(t))\mathbf{C}^{\mathrm{T}}(t, \boldsymbol{x}(t), \boldsymbol{u}(t))\big)\mathrm{d}t \right] \\
&= \inf_{\boldsymbol{u} \in \mathcal{U}} \left[ \left(q(\boldsymbol{x}(t)) + \frac{1}{2}\boldsymbol{u}(t)^{\mathrm{T}}\mathbf{R}\boldsymbol{u}(t)\right)\mathrm{d}t + V(t, \boldsymbol{x}(t)) + V_t(t, \boldsymbol{x}(t))\mathrm{d}t \right. \\
&\quad + V_{\boldsymbol{x}}^{\mathrm{T}}(t, \boldsymbol{x}(t))\Big(\boldsymbol{f}(t, \boldsymbol{x}(t))\mathrm{d}t + \mathbf{G}(t, \boldsymbol{x}(t))\boldsymbol{u}(t)\mathrm{d}t\Big) \\
&\quad \left. + \frac{1}{2}\mathrm{tr}\big(V_{\boldsymbol{x}\boldsymbol{x}}(t, \boldsymbol{x}(t))\mathbf{C}(t, \boldsymbol{x}(t), \boldsymbol{u}(t))\mathbf{C}^{\mathrm{T}}(t, \boldsymbol{x}(t), \boldsymbol{u}(t))\big) \right]\mathrm{d}t
\end{aligned}
$$

We can cancel $V(t, \boldsymbol{x}(t))$ on both sides and bring the terms not dependent on controls outside of the infimum to get

$$V_t + V_{\boldsymbol{x}}^{\mathrm{T}}\boldsymbol{f} + q + \inf_{\boldsymbol{u} \in \mathcal{U}[0, T]} \left\{ \frac{1}{2}\boldsymbol{u}^{\mathrm{T}}R\boldsymbol{u} + V_{\boldsymbol{x}}^{\mathrm{T}}\mathbf{G}\boldsymbol{u} + \frac{1}{2}\mathrm{tr}(V_{\boldsymbol{x}\boldsymbol{x}}\mathbf{C}\mathbf{C}^{\mathrm{T}}) \right\} = 0 \tag{15}$$

Note that the functional dependencies have been dropped for brevity. Now we can find the optimal control by taking the gradient of the terms inside the infimum with respect to $\boldsymbol{u}$ and setting it to zero. Before that we first substitute for $\mathbf{C} = [\sigma \mathbf{G}\boldsymbol{u}, \ \boldsymbol{\Sigma}]$ so that the term inside the trace operator becomes $\sigma^2 V_{\boldsymbol{x}\boldsymbol{x}}\mathbf{G}\boldsymbol{u}\boldsymbol{u}^{\mathrm{T}}\mathbf{G}^{\mathrm{T}} + V_{\boldsymbol{x}\boldsymbol{x}}\boldsymbol{\Sigma}\boldsymbol{\Sigma}^{\mathrm{T}}$. By using the property of trace operators, we can write, $\mathrm{tr}(V_{\boldsymbol{x}\boldsymbol{x}}\mathbf{C}\mathbf{C}^{\mathrm{T}}) = \mathrm{tr}(\sigma^2 V_{\boldsymbol{x}\boldsymbol{x}}\mathbf{G}\boldsymbol{u}\boldsymbol{u}^{\mathrm{T}}\mathbf{G}^{\mathrm{T}} + V_{\boldsymbol{x}\boldsymbol{x}}\boldsymbol{\Sigma}\boldsymbol{\Sigma}^{\mathrm{T}}) = \mathrm{tr}(\sigma^2 \boldsymbol{u}^{\mathrm{T}}\mathbf{G}^{\mathrm{T}}V_{\boldsymbol{x}\boldsymbol{x}}\mathbf{G}\boldsymbol{u}) + \mathrm{tr}(V_{\boldsymbol{x}\boldsymbol{x}}\boldsymbol{\Sigma}\boldsymbol{\Sigma}^{\mathrm{T}})$. Note that the second term is not a function of $\boldsymbol{u}$ and therefore is zero when we take the gradient. Additionally, the first term is a scalar and taking its trace returns the same scalar. Now, taking the gradient w.r.t $\boldsymbol{u}$, we get,

$$
\begin{aligned}
\mathbf{R}\boldsymbol{u}^* + (V_{\boldsymbol{x}}^{\mathrm{T}}\mathbf{G})^{\mathrm{T}} + \sigma^2 \mathbf{G}^{\mathrm{T}}V_{\boldsymbol{x}\boldsymbol{x}}\mathbf{G}\boldsymbol{u}^* &= 0 \\
\Rightarrow (\mathbf{R} + \sigma^2 \mathbf{G}^{\mathrm{T}}V_{\boldsymbol{x}\boldsymbol{x}}\mathbf{G})\boldsymbol{u}^* &= -(\mathbf{G}^{\mathrm{T}}V_{\boldsymbol{x}}) \\
\Rightarrow \hat{\mathbf{R}}\boldsymbol{u}^* &= -(\mathbf{G}^{\mathrm{T}}V_{\boldsymbol{x}}) \\
\Rightarrow \boldsymbol{u}^* &= -\hat{\mathbf{R}}^{-1}\mathbf{G}^{\mathrm{T}}V_{\boldsymbol{x}}.
\end{aligned}
$$

where, $\hat{\mathbf{R}} \triangleq (\mathbf{R} + \sigma^2 \mathbf{G}^{\mathrm{T}}V_{\boldsymbol{x}\boldsymbol{x}}\mathbf{G})$.

The optimal control above can be plugged back into equation 15 to obtain

$$V_t + V_{\boldsymbol{x}}^{\mathrm{T}}\boldsymbol{f} + q + \frac{1}{2}V_{\boldsymbol{x}}^{\mathrm{T}}\mathbf{G}\hat{\mathbf{R}}^{-\mathrm{T}}\mathbf{R}\hat{\mathbf{R}}^{-1}\mathbf{G}^{\mathrm{T}}V_{\boldsymbol{x}} - V_{\boldsymbol{x}}^{\mathrm{T}}\mathbf{G}\hat{\mathbf{R}}^{-1}\mathbf{G}^{\mathrm{T}}V_{\boldsymbol{x}}$$

$$+ \frac{1}{2}\sigma^2 V_{\boldsymbol{x}}^{\mathrm{T}}\mathbf{G}\hat{\mathbf{R}}^{-\mathrm{T}}\mathbf{G}^{\mathrm{T}}V_{\boldsymbol{x}\boldsymbol{x}}\mathbf{G}\hat{\mathbf{R}}^{-1}\mathbf{G}^{\mathrm{T}}V_{\boldsymbol{x}} + \frac{1}{2}\operatorname{tr}(V_{\boldsymbol{x}\boldsymbol{x}}\boldsymbol{\Sigma}\boldsymbol{\Sigma}^{\mathrm{T}}) = 0$$

$$\Rightarrow V_t + V_{\boldsymbol{x}}^{\mathrm{T}}\boldsymbol{f} + q + \frac{1}{2}V_{\boldsymbol{x}}^{\mathrm{T}}\mathbf{G}\hat{\mathbf{R}}^{-\mathrm{T}}(\mathbf{R} + \sigma^2\mathbf{G}^{\mathrm{T}}V_{\boldsymbol{x}\boldsymbol{x}}\mathbf{G})\hat{\mathbf{R}}^{-1}\mathbf{G}^{\mathrm{T}}V_{\boldsymbol{x}}$$

$$- V_{\boldsymbol{x}}^{\mathrm{T}}\mathbf{G}\hat{\mathbf{R}}^{-1}\mathbf{G}^{\mathrm{T}}V_{\boldsymbol{x}} + \frac{1}{2}\operatorname{tr}(V_{\boldsymbol{x}\boldsymbol{x}}\boldsymbol{\Sigma}\boldsymbol{\Sigma}^{\mathrm{T}}) = 0$$

$$\Rightarrow V_t + V_{\boldsymbol{x}}^{\mathrm{T}}\boldsymbol{f} + q + \frac{1}{2}V_{\boldsymbol{x}}^{\mathrm{T}}\mathbf{G}\hat{\mathbf{R}}^{-\mathrm{T}}\hat{\mathbf{R}}\hat{\mathbf{R}}^{-1}\mathbf{G}^{\mathrm{T}}V_{\boldsymbol{x}}$$

$$- V_{\boldsymbol{x}}^{\mathrm{T}}\mathbf{G}\hat{\mathbf{R}}^{-1}\mathbf{G}^{\mathrm{T}}V_{\boldsymbol{x}} + \frac{1}{2}\operatorname{tr}(V_{\boldsymbol{x}\boldsymbol{x}}\boldsymbol{\Sigma}\boldsymbol{\Sigma}^{\mathrm{T}}) = 0$$

$$\Rightarrow V_t + V_{\boldsymbol{x}}^{\mathrm{T}}\boldsymbol{f} + q - \frac{1}{2}V_{\boldsymbol{x}}^{\mathrm{T}}\mathbf{G}\hat{\mathbf{R}}^{-1}\mathbf{G}^{\mathrm{T}}V_{\boldsymbol{x}} + \frac{1}{2}\operatorname{tr}(V_{\boldsymbol{x}\boldsymbol{x}}\boldsymbol{\Sigma}\boldsymbol{\Sigma}^{\mathrm{T}}) = 0$$

which gives the HJB PDE as mentioned in equation 4.

## C  DERIVATION OF THE 2FBSDES

Firstly we can apply Ito's differentiation rule to $V$:

$$dV \overset{\text{Itô}}{=} V_t dt + V_{\boldsymbol{x}}^{\mathrm{T}} d\boldsymbol{x} + \frac{1}{2}\operatorname{tr}\left(V_{\boldsymbol{x}\boldsymbol{x}}\mathbf{C}\mathbf{C}^{\mathrm{T}}\right)dt. \tag{16}$$

Since $V$ is the value function, we can substitute in the HJB PDE for $V_t$ and forward dynamics for $d\boldsymbol{x}$ to get:

$$dV = -\Big(q + V_{\boldsymbol{x}}^{\mathrm{T}}\boldsymbol{f} - \frac{1}{2}V_{\boldsymbol{x}}^{\mathrm{T}}\mathbf{G}\hat{\mathbf{R}}^{-1}\mathbf{G}^{\mathrm{T}}V_{\boldsymbol{x}} + \frac{1}{2}\operatorname{tr}\left(V_{\boldsymbol{x}\boldsymbol{x}}\boldsymbol{\Sigma}\boldsymbol{\Sigma}^{\mathrm{T}}\right)\Big)dt$$

$$+ V_{\boldsymbol{x}}^{\mathrm{T}}\Big(\boldsymbol{f}dt + \mathbf{G}(\boldsymbol{u}^*dt + \sigma\boldsymbol{u}^*dv) + \boldsymbol{\Sigma}d\boldsymbol{w}\Big) + \frac{1}{2}\operatorname{tr}\left(V_{\boldsymbol{x}\boldsymbol{x}}\mathbf{C}\mathbf{C}^{\mathrm{T}}\right)dt$$

$$= -\Big(q + V_{\boldsymbol{x}}^{\mathrm{T}}\boldsymbol{f} - \frac{1}{2}V_{\boldsymbol{x}}^{\mathrm{T}}\mathbf{G}\hat{\mathbf{R}}^{-1}\mathbf{G}^{\mathrm{T}}V_{\boldsymbol{x}} + \frac{1}{2}\operatorname{tr}\left(V_{\boldsymbol{x}\boldsymbol{x}}\boldsymbol{\Sigma}\boldsymbol{\Sigma}^{\mathrm{T}}\right)\Big)dt$$

$$+ V_{\boldsymbol{x}}^{\mathrm{T}}\Big(\boldsymbol{f}dt + \mathbf{G}(\boldsymbol{u}^*dt + \sigma\boldsymbol{u}^*dv) + \boldsymbol{\Sigma}d\boldsymbol{w}\Big) + \frac{1}{2}\operatorname{tr}\left(V_{\boldsymbol{x}\boldsymbol{x}}\boldsymbol{\Sigma}\boldsymbol{\Sigma}^{\mathrm{T}} + \sigma^2\boldsymbol{u}^{*\mathrm{T}}\mathbf{G}^{\mathrm{T}}V_{\boldsymbol{x}\boldsymbol{x}}\mathbf{G}\boldsymbol{u}^*\right)dt$$

$$= -\Big(q + V_{\boldsymbol{x}}^{\mathrm{T}}\boldsymbol{f} - \frac{1}{2}V_{\boldsymbol{x}}^{\mathrm{T}}\mathbf{G}\hat{\mathbf{R}}^{-1}\mathbf{G}^{\mathrm{T}}V_{\boldsymbol{x}} - \frac{1}{2}\sigma^2 V_{\boldsymbol{x}}^{\mathrm{T}}\mathbf{G}\hat{\mathbf{R}}^{-\mathrm{T}}\mathbf{G}^{\mathrm{T}}V_{\boldsymbol{x}\boldsymbol{x}}\mathbf{G}\hat{\mathbf{R}}^{-1}\mathbf{G}^{\mathrm{T}}V_{\boldsymbol{x}}\Big)dt$$

$$+ V_{\boldsymbol{x}}^{\mathrm{T}}\Big(\boldsymbol{f}dt + \mathbf{G}(\boldsymbol{u}^*dt + \sigma\boldsymbol{u}^*dv) + \boldsymbol{\Sigma}d\boldsymbol{w}\Big)$$

$$= -\Big(q + V_{\boldsymbol{x}}^{\mathrm{T}}\boldsymbol{f} - \frac{1}{2}V_{\boldsymbol{x}}^{\mathrm{T}}\mathbf{G}\hat{\mathbf{R}}^{-\mathrm{T}}\hat{\mathbf{R}}\hat{\mathbf{R}}^{-1}\mathbf{G}^{\mathrm{T}}V_{\boldsymbol{x}} - \frac{1}{2}\sigma^2 V_{\boldsymbol{x}}^{\mathrm{T}}\mathbf{G}\hat{\mathbf{R}}^{-\mathrm{T}}\mathbf{G}^{\mathrm{T}}V_{\boldsymbol{x}\boldsymbol{x}}\mathbf{G}\hat{\mathbf{R}}^{-1}\mathbf{G}^{\mathrm{T}}V_{\boldsymbol{x}}\Big)dt$$

$$+ V_{\boldsymbol{x}}^{\mathrm{T}}\Big(\boldsymbol{f}dt + \mathbf{G}(\boldsymbol{u}^*dt + \sigma\boldsymbol{u}^*dv) + \boldsymbol{\Sigma}d\boldsymbol{w}\Big)$$

$$= -\Big(q + V_{\boldsymbol{x}}^{\mathrm{T}}\boldsymbol{f} - \frac{1}{2}V_{\boldsymbol{x}}^{\mathrm{T}}\mathbf{G}\hat{\mathbf{R}}^{-\mathrm{T}}(\mathbf{R} + 2\sigma^2\mathbf{G}^{\mathrm{T}}V_{\boldsymbol{x}\boldsymbol{x}}\mathbf{G})\hat{\mathbf{R}}^{-1}\mathbf{G}^{\mathrm{T}}V_{\boldsymbol{x}}\Big)dt$$

$$+ V_{\boldsymbol{x}}^{\mathrm{T}}\Big(\boldsymbol{f}dt + \mathbf{G}(\boldsymbol{u}^*dt + \sigma\boldsymbol{u}^*dv) + \boldsymbol{\Sigma}d\boldsymbol{w}\Big)$$

$$= -(q - \frac{1}{2}V_{\boldsymbol{x}}^{\mathrm{T}}\mathbf{G}\hat{\mathbf{R}}^{-\mathrm{T}}(\mathbf{R} + 2\sigma^2\mathbf{G}^{\mathrm{T}}V_{\boldsymbol{x}\boldsymbol{x}}\mathbf{G})\hat{\mathbf{R}}^{-1}\mathbf{G}^{\mathrm{T}}V_{\boldsymbol{x}}\Big)dt + V_{\boldsymbol{x}}^{\mathrm{T}}\mathbf{G}(\boldsymbol{u}^*dt + \sigma\boldsymbol{u}^*dv) + V_{\boldsymbol{x}}^{\mathrm{T}}\boldsymbol{\Sigma}d\boldsymbol{w}. \tag{17}$$

For $V_{\boldsymbol{x}}$, we can again apply Ito's differentiation rule:

$$dV_{\boldsymbol{x}} \overset{\text{Itô}}{=} V_{\boldsymbol{x}t}dt + V_{\boldsymbol{x}\boldsymbol{x}}^{\mathrm{T}}d\boldsymbol{x} + \frac{1}{2}\operatorname{tr}\left(V_{\boldsymbol{x}\boldsymbol{x}\boldsymbol{x}}\mathbf{C}\mathbf{C}^{\mathrm{T}}\right). \tag{18}$$

We can again substitute in the forward dynamics for $d\boldsymbol{x}$ and get:

$$dV_{\boldsymbol{x}} = \partial_t V_{\boldsymbol{x}}dt \tag{19}$$

$$+ \partial_{\boldsymbol{x}} V_{\boldsymbol{x}}^{\mathrm{T}}\Big(\boldsymbol{f}dt + \mathbf{G}(\boldsymbol{u}^*dt + \sigma\boldsymbol{u}^*dv) + \boldsymbol{\Sigma}d\boldsymbol{w}\Big)$$

$$+ \frac{1}{2}\operatorname{tr}\left(\partial_{\boldsymbol{x}\boldsymbol{x}} V_{\boldsymbol{x}}\mathbf{C}\mathbf{C}^{\mathrm{T}}\right)dt$$

$$= (\mathbf{A} + V_{\boldsymbol{x}\boldsymbol{x}}\boldsymbol{f})dt + V_{\boldsymbol{x}\boldsymbol{x}}\mathbf{G}(\boldsymbol{u}^*dt + \sigma\boldsymbol{u}^*dv) \tag{20}$$

$$+ V_{\boldsymbol{x}\boldsymbol{x}}\boldsymbol{\Sigma}d\boldsymbol{w}. \tag{21}$$

We have $\mathbf{A} = \partial_t V_{\boldsymbol{x}} + \frac{1}{2} \operatorname{tr}(\partial_{\boldsymbol{x}\boldsymbol{x}} V_{\boldsymbol{x}} \mathbf{C}\mathbf{C}^{\mathrm{T}})$. Note that the transpose on $V_{\boldsymbol{x}\boldsymbol{x}}$ is dropped since it is symmetric.

## D  LOSS FUNCTION

The loss function used in this work builds on the loss functions used in Han et al. (2018) and Pereira et al. (2019). Because the Deep 2FBSDE Controller propagates 2 BSDEs, in addition to using the propagated value function $(V_t)$, the propagated gradient of the value function $(V_{\boldsymbol{x},t})$ can also be used in the loss function to enforce that the network meets both the terminal constraints i.e. $\phi(\boldsymbol{x}_N)$ and $\phi_{\boldsymbol{x}}(\boldsymbol{x}_N)$ respectively. Moreover, since the terminal cost function is known, its Hessian can be computed and used to enforce that the output of the $FC_1$ layer at the terminal time index $(V_{\boldsymbol{x}\boldsymbol{x},N})$ be equal to the target Hessian of the terminal cost function $\phi_{\boldsymbol{x}\boldsymbol{x}}(\boldsymbol{x}_N)$. Although this is enforced only at the terminal time index $(V_{\boldsymbol{x}\boldsymbol{x},N})$, because the weights of a recurrent neural network are shared across time, in order to be able to predict $V_{\boldsymbol{x}\boldsymbol{x},N}$ accurately all of the prior predictions $V_{\boldsymbol{x}\boldsymbol{x},t}$ will have to be adjusted accordingly, thereby representing some form of propagation of the Hessian of the value function.

Additionally, applying the optimal control, which uses the network prediction, to the forward process introduces an additional gradient path through the system dynamics at every time step. Although this makes training difficult (gradient vanishing problem) it allows the weights to now influence what the next state (i.e. at the next time index) will be. As a result, the weights can control the state at the end time index and hence the target $(V^*(\boldsymbol{x}_N))$ for the neural network prediction itself. This can be added to the loss function which can accelerate the minimization of the terminal cost to achieve the task objectives. The final form of our loss function is

$$\mathcal{L} = c_1 \, ||V^* - V_N||_2^2 + c_2 \, ||V_{\boldsymbol{x}}^* - V_{\boldsymbol{x},N}||_2^2 + c_3 \, ||V_{\boldsymbol{x}\boldsymbol{x}}^* - V_{\boldsymbol{x}\boldsymbol{x},N}||_2^2 + c_4 \, ||V^*||_2^2 + \lambda ||\theta||_2^2,$$

where $V^* = \phi(\boldsymbol{x}_N)$, $V_{\boldsymbol{x}}^* = \phi_{\boldsymbol{x}}(\boldsymbol{x}_N)$ and $V_{\boldsymbol{x}\boldsymbol{x}}^* = \phi_{\boldsymbol{x}\boldsymbol{x}}(\boldsymbol{x}_N)$.

## E  NON-LINEAR SYSTEM DYNAMICS AND SYSTEM PARAMETERS

For all simulation experiments, we used the same quadratic state cost of the form $q(\boldsymbol{x}) = \frac{1}{2}(\boldsymbol{x} - \boldsymbol{x}_{\text{target}})^{\mathrm{T}} \mathbf{Q}(\boldsymbol{x} - \boldsymbol{x}_{\text{target}})$ for both the running and terminal state costs.

### E.1  CARTPOLE

The cartpole dynamics is given by

$$\mathrm{d}\begin{bmatrix} x \\ \theta \\ \dot{x} \\ \dot{\theta} \end{bmatrix} = \begin{bmatrix} \dot{x} \\ \dot{\theta} \\ \frac{m_p \sin\theta(l\dot{\theta}^2 + g\cos\theta)}{m_c + m_p \sin^2\theta} \\ \frac{-m_p l\dot{\theta}^2 \cos\theta \sin\theta - (m_c + m_p)g\sin\theta}{l(m_c + m_p \sin^2\theta)} \end{bmatrix} \mathrm{d}t + \begin{bmatrix} 0 \\ 0 \\ \frac{1}{m_c + m_p \sin^2\theta} \\ \frac{-1}{l(m_c + m_p \sin^2\theta)} \end{bmatrix}(u\mathrm{d}t + \sigma u\mathrm{d}v) + \begin{bmatrix} \mathbf{0}_{2\times 2} & \mathbf{0}_{2\times 2} \\ \mathbf{0}_{2\times 2} & \mathbf{I}_{2\times 2} \end{bmatrix}\mathrm{d}\boldsymbol{w}.$$

The model and dynamics parameters are set as $m_p = 0.01$ kg, $m_c = 1$ kg, $l = 0.5$ m, $\sigma = 0.125$. The initial pole and cart position are 0 rad and 0 m with no velocity, and the target state is a pole angle of $\pi$ rad and zeros for all other states. The control costs $\mathbf{R}$ are 0.5 and 0.1 for experiments in fig.3 and fig. 6 respectively. The state cost matrix is the same for both experiments at $\mathbf{Q} = diag\,[0.0, 6.0, 0.3, 0.3]$.

### E.2  QUADCOPTER

The quadcopter dynamics used can be found in ElKholy (2014). Its dynamics is given by

$$d\begin{bmatrix} x \\ y \\ z \\ \phi \\ \theta \\ \psi \\ \dot{x} \\ \dot{y} \\ \dot{z} \\ \dot{\phi} \\ \dot{\theta} \\ \dot{\psi} \end{bmatrix} = \begin{bmatrix} \dot{x} \\ \dot{y} \\ \dot{z} \\ \dot{\phi} \\ \dot{\theta} \\ \dot{\psi} \\ 0 \\ 0 \\ g \\ \frac{I_{yy}}{I_{xx}}\dot{\psi}\dot{\theta} - \frac{I_{zz}}{I_{xx}}\dot{\theta}\dot{\psi} \\ \frac{I_{zz}}{I_{yy}}\dot{\phi}\dot{\psi} - \frac{I_{xx}}{I_{yy}}\dot{\psi}\dot{\phi} \\ \frac{I_{xx}}{I_{zz}}\dot{\theta}\dot{\phi} - \frac{I_{yy}}{I_{zz}}\dot{\phi}\dot{\theta} \end{bmatrix} dt + \begin{bmatrix} 0 & 0 & 0 & 0 \\ 0 & 0 & 0 & 0 \\ 0 & 0 & 0 & 0 \\ 0 & 0 & 0 & 0 \\ 0 & 0 & 0 & 0 \\ 0 & 0 & 0 & 0 \\ -\frac{1}{m}(\sin\phi\sin\psi + \cos\phi\cos\psi\sin\theta) & 0 & 0 & 0 \\ -\frac{1}{m}(\cos\phi\sin\psi\sin\theta - \cos\psi\sin\phi) & 0 & 0 & 0 \\ -\frac{1}{m}\cos\phi\cos\theta & 0 & 0 & 0 \\ 0 & \frac{l}{I_{xx}} & 0 & 0 \\ 0 & 0 & \frac{l}{I_{yy}} & 0 \\ 0 & 0 & 0 & \frac{1}{I_{zz}} \end{bmatrix} \left( \begin{bmatrix} u_1 \\ u_2 \\ u_3 \\ u_4 \end{bmatrix} dt \right.$$

$$+ \sigma \begin{bmatrix} u_1 \\ u_2 \\ u_3 \\ u_4 \end{bmatrix} dv \right) + \begin{bmatrix} \mathbf{0}_{6\times 6} & \mathbf{0}_{6\times 6} \\ \mathbf{0}_{6\times 6} & \mathbf{I}_{6\times 6} \end{bmatrix} d\boldsymbol{w},$$

where

$$\begin{bmatrix} u_1 \\ u_2 \\ u_3 \\ u_4 \end{bmatrix} = \begin{bmatrix} 1 & 1 & 1 & 1 \\ 0 & -1 & 0 & 1 \\ 1 & 0 & -1 & 0 \\ d & -d & d & -d \end{bmatrix} \begin{bmatrix} \tau_1 \\ \tau_2 \\ \tau_3 \\ \tau_4 \end{bmatrix}.$$

The states are positions, velocities, angles and angular rates, and the controls are the four rotor torques ($\tau$). The model and dynamics parameters are set as $m = 0.47$ kg, $I_{xx} = I_{yy} = 4.86 \times 10^{-3}$ kg m$^2$, $I_{zz} = 8.8 \times 10^{-3}$kgm$^2$, $l = 0.225$ m, $d = 0.05$. The initial state conditions used are all zeros, and the target state is 1 meter each in the north-east-down directions and all other states zero. The control cost matrix is $\mathbf{R} = 2\mathbf{I}_{4\times 4}$. The state cost matrix is $\mathbf{Q} = diag[20.0, 20.0, 50.0, 5.0, 5.0, 5.0, 1.25, 1.25, 5.0, 0.25, 0.25, 0.25]$.

### E.3 2-LINK 6-MUSCLE HUMAN ARM

The forward rigid-body dynamics of the 2-link human arm as stated in Li & Todorov (2004) is given by:

$$\ddot{\boldsymbol{\theta}} = \mathcal{M}(\boldsymbol{\theta})^{-1}(\boldsymbol{\tau} - \mathcal{C}(\boldsymbol{\theta}, \dot{\boldsymbol{\theta}}) - \mathcal{B}\dot{\boldsymbol{\theta}})$$

where $\theta \in \mathbb{R}^2$ represents the joint angle vector consisting of $\theta_1$ (shoulder joint angle) and $\theta_2$ (elbow joint angle). $\mathcal{M}(\theta) \in \mathbb{R}^{2\times 2}$ is the inertia matrix which is positive definite and symmetric. $\mathcal{C}(\theta, \dot{\theta}) \in \mathbb{R}^2$ is the vector centripetal and Coriolis forces. $\mathcal{B} \in \mathbb{R}^{2\times 2}$ is the joint friction matrix. $\tau \in \mathbb{R}^2$ is the joint torque applied on the system. The torque is generated by activation of 6 muscle groups. Following are equations for components of each of the above matrices and vectors:

$$\mathcal{M} = \begin{pmatrix} \tilde{a}_1 + 2\tilde{a}_2 cos\theta_2 & \tilde{a}_3 + \tilde{a}_2 cos\theta_2 \\ \tilde{a}_3 + \tilde{a}_2 cos\theta_2 & \tilde{a}_3 \end{pmatrix}$$

$$\mathcal{C} = \begin{pmatrix} -\dot{\theta}_2(2\dot{\theta}_1 + \dot{\theta}_2) \\ \dot{\theta}_1 \end{pmatrix} a_2 sin\theta_2$$

$$\mathcal{B} = \begin{pmatrix} b_{11} & b_{12} \\ b_{21} & b_{22} \end{pmatrix}$$

$$\tilde{a}_1 = I_1 + I_2 + M_2 l_1^2$$

$$\tilde{a}_2 = m_2 L_1 s_2$$

$$\tilde{a}_3 = I_2$$

Where $b_{11} = b_{22} = 0.05$, $b_{12} = b21 = 0.025$, $m_i$ is the mass (1.4kg,1kg), $l_i$ is the length of link i (30cm, 33cm), $s_i$ is the distance between joint center and mass of link $i$ (11cm, 16cm), $I_i$ represents for the moment of inertia ($0.025kgm^2$,$0.045kgm^2$).
The activation dynamics for each muscle is given by:

$$\dot{a}_i = \frac{(1 + \sigma dv)u_i - a_i}{\tilde{t}}, \forall i \in [1, 6]$$

$$\tilde{t} = t_{deact}$$

where $t_{deact} = 66msec$, $a \in \mathbb{R}^6$ is a vector of muscle activations, $u \in \mathbb{R}^6$ is a vector of instantaneous neural inputs. As a result of these dynamics, six new state variables ($\boldsymbol{a}$) will be incorporated into the dynamical system. With the muscle activation dynamics proposed above, the joint torque vector can be computed as follows:

$$\boldsymbol{\tau} = M(\boldsymbol{\theta})T(a, l(\boldsymbol{\theta}), v(\boldsymbol{\theta}, \dot{\boldsymbol{\theta}}))$$

wherein $T(a, l, v) \in \mathbb{R}$ is the tensile force generated by each muscle, whose expression is given by:

$$T(a, l, v) = A(a, l)(F_l(l)F_V(l, v) + F_P(l))$$

$$A(a, l) = 1 - \exp\left(-\left(\frac{a}{0.56N_f(l)}\right)^{N_f(l)}\right)$$

$$N_f(l) = 2.11 + 4.16\left(\frac{1}{l} - 1\right)$$

$$F_L(l) = \exp\left(-\left|\frac{l^{1.93} - 1}{1.03}\right|^{1.87}\right)$$

$$F_V(l, v) = \begin{cases} \frac{-5.72 - v}{-5.72 + (1.38 + 2.09l)v}, & \text{if } v \leq 0 \\ \frac{0.62 - (-3.12 + 4.21l - 2.67l^2)v}{0.62 + v}, & \text{otherwise} \end{cases}$$

$$F_p(l) = -0.02 \exp(13.8 - 18.7l)$$

The equation for $M(\theta)$ is obtained from Teka et al. (2017, Equation 3). The parameters used in the equations above are mostly obtained by work in Teka et al. (2017), which we provide here in the following table:

| Muscle | Maximal Force, $F_{max}$ (N) | Optimal Length, $L_{opt}$ (m) | Velocity, $v$ (m/sec) |
|--------|------------------------------|-------------------------------|------------------------|
| SF | 420 | 0.185 | $-\dot{\theta}_1 R_{SF}$ |
| SE | 570 | 0.170 | $-\dot{\theta}_1 R_{SE}$ |
| EF | 1010 | 0.180 | $-(\dot{\theta}2 - \dot{\theta}_1)R_{EF}$ |
| EE | 1880 | 0.055 | $-(\dot{\theta}2 - \dot{\theta}_1)R_{EE}$ |
| BF | 460 | 0.130 | $-\dot{\theta}_1 R_{BFS} - (\dot{\theta}_2 - \dot{\theta}_1)R_{BFE}$ |
| BE | 630 | 0.150 | $-\dot{\theta}_1 R_{BFS} - (\dot{\theta}_2 - \dot{\theta}_1)R_{BEE}$ |

Where the last column consists of moment arms which are approximated according to (Li & Todorov, 2004, figure B). $l \in \mathbb{R}^{6 \times 1}$ is the length of each muscle, which is calculated by approximating the range of each muscle group in Li & Todorov (2004) using the data of joint angle ranges and linear relationships with corresponding joint angles as in Teka et al. (2017). Each of the fitted linear functions for muscle lengths and cosine functions for moment arms are available in the provided MATLAB code.

The above system is a deterministic model of the 2-link 6-muscle human arm. However, in the main paper we consider experiments with both additive and control multiplicative stochasticities. A state-space SDE model for

the human arm is given as follows,

$$
d\begin{bmatrix} \theta_1 \\ \theta_2 \\ \dot{\theta}_1 \\ \dot{\theta}_2 \\ a_1 \\ a_2 \\ a_3 \\ a_4 \\ a_5 \\ a_6 \end{bmatrix} = \begin{bmatrix} \dot{\theta}_1 \\ \dot{\theta}_2 \\ \ddot{\theta}_1 \\ \ddot{\theta}_2 \\ \frac{-a_1}{t_{deact}} \\ \frac{-a_2}{t_{deact}} \\ \frac{-a_3}{t_{deact}} \\ \frac{-a_4}{t_{deact}} \\ \frac{-a_5}{t_{deact}} \\ \frac{-a_6}{t_{deact}} \end{bmatrix} dt + \begin{bmatrix} \mathbf{0}_{4\times 2} \\ \frac{1}{t_{deact}}\mathbf{I}_{6\times 6} \end{bmatrix}(\boldsymbol{u}dt + \sigma\boldsymbol{u}dv) + \begin{bmatrix} \mathbf{0}_{2\times 2} \\ \boldsymbol{\Sigma}_{2\times 2} \\ \mathbf{0}_{6\times 2} \end{bmatrix}d\boldsymbol{w}.
$$

where, $\ddot{\boldsymbol{\theta}} = [\ddot{\theta}_1, \ \ddot{\theta}_2]^{\mathrm{T}} = \mathcal{M}(\boldsymbol{\theta})^{-1}(\boldsymbol{\tau} - \mathcal{C}(\boldsymbol{\theta}, \dot{\boldsymbol{\theta}}) - \mathcal{B}\dot{\boldsymbol{\theta}})$ and $\boldsymbol{u} = [u_1, \ u_2, \ u_3, \ u_4, \ u_5, \ u_6]^{\mathrm{T}}$

In this system, the control cost coefficient matrix is $\mathbf{R} = 2\mathbf{I}_{6\times 6}$. The state cost matrix is $\mathbf{Q} = diag[10.0, 10.0, 0.1, 0.1, 0.001, 0.001, 0.001, 0.001, 0.001, 0.001]$.

## F  SIMULATION PARAMETERS

The precise training/network parameters are shown in the following table

| system | batch size | iterations | $c_1$ | $c_2$ | $c_3$ | $c_4$ | $\lambda$ | d$t$ (sec) | $T$ (sec) | $h$ |
|---|---|---|---|---|---|---|---|---|---|---|
| CartPole | 256 | 2000 | 1 | 1 | 1 | 1 | 0.0005 | 0.02 | 2 | $[8, 8]$ |
| QuadCopter | 256 | 5000 | 1 | 1 | 1 | 1 | 0.0005 | 0.02 | 2 | $[8, 8]$ |
| HumanHand | 256 | 2000 | 1 | 0 | 1 | 1 | 0.0005 | 0.02 | 2 | $[8, 8]$ |

where iterations is the number of epochs. $c_1$ to $c_4$ represent of the weight of the components in the loss function in SM D. $\lambda$ is the weights for $L_2$ regularization of neural network. d$t$ is the time discretization. $T$ is the time horizon for the task. $h$ is the output size of each LSTM layer (cell state dimension is the same).

## G  ADDITIONAL PLOTS

### G.1  CARTPOLE SWING-UP WITH LOWER CONTROL COST

The plots in fig. 6 show the comparison of the 2FBSDE, FBSDE and iLQG controllers when the restriction on the controls is lowered as compared to that in the main paper. Here, the control cost is set to $R = 0.1$ as compared to $R = 0.5$ in the main paper. Clearly, the behavior is very different. Now, all three controllers try to reach the targets as soon as possible by applying the required control effort thus ignoring the presence of control multiplicative noise in the system.

### G.2  2-LINK 6-MUSCLE HUMAN ARM SINGLE CONTROL TRAJECTORY SAMPLE

Fig. 7 includes a single control trajectory as well as the mean and variance for the human arm simulation to demonstrate the feasibility of control trajectories from 2FBSDE.

### G.3  QUADCOPTER VELOCITY STATES

Fig. 8 shows the velocity trajectories of the quadcopter experiment in the main paper.

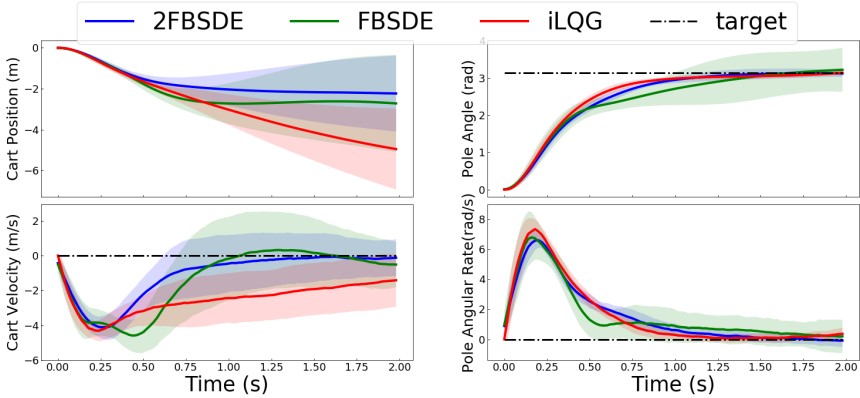

Figure 6: Comparison of 2FBSDE, FBSDE and iLQG on Cartpole with control cost of $0.1$.

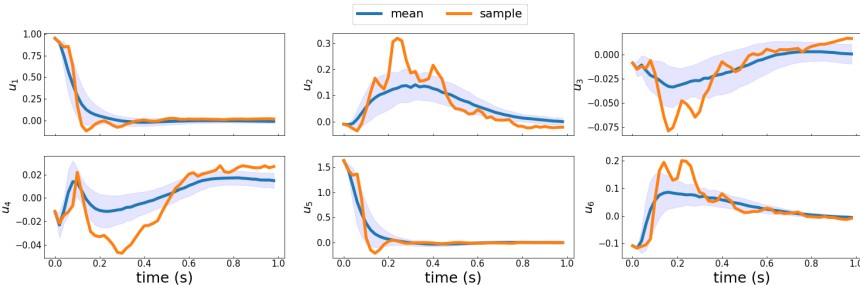

Figure 7: Single control trajectory as well as mean and variance of 2FBSDE on Human Arm. The single trajectory demonstrates the feasibility of the resulting controls.

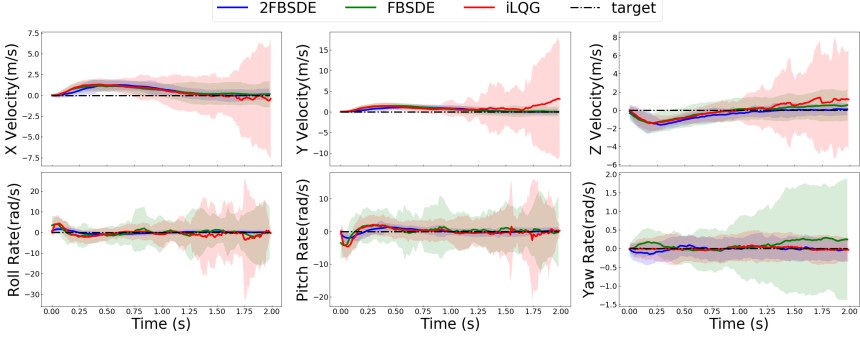

Figure 8: Comparison of 2FBSDE, FBSDE and iLQG on the quadcopter velocities and angular rates. Again the variance of 2FBSDE trajectories are smaller than FBSDE and iLQG.

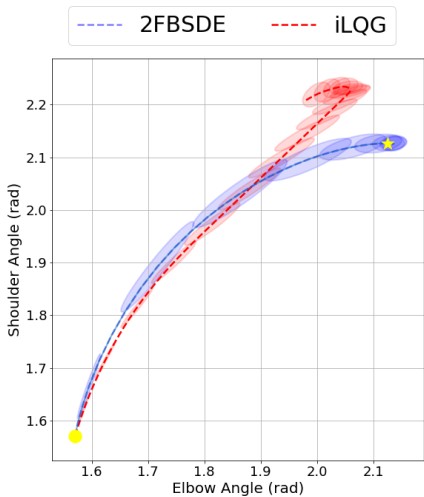

Figure 9: Phase plot at control multiplicative standard deviation $\sigma = 0.1$ and additive noise standard deviation $\Sigma = 0.1\mathbf{I}$. The yellow circle denotes the starting point, and yellow star denotes the target.

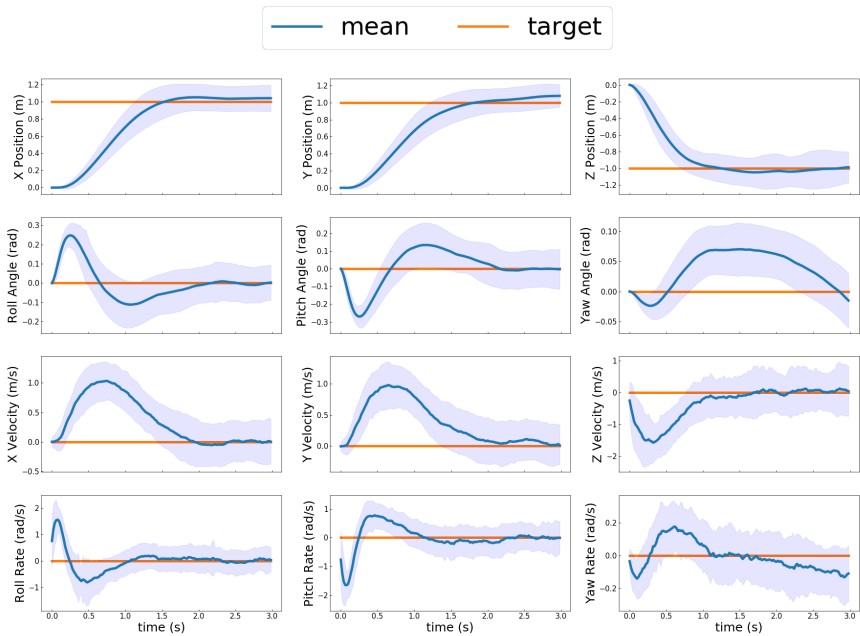

Figure 10: Quadcopter state trajectories of 2FBSDE for 3 seconds. The plots demonstrate the controller's capability to reach and hover.

### G.4 QUADCOPTER LONGER TIME HORIZON

Fig. 10 shows quadcopter state trajectories of 2FBSDE for longer time horizon (3 seconds).

### G.5 PHASE PLOT FOR HUMAN ARM SIMULATION

Fig. 9 is a phase plot of the human arm experiment experiment at $\sigma = 0.1$ and $\Sigma = 0.1\mathbf{I}$. It clearly shows that 2FBSDE can reach the target location and maintain small variance in the trajectory whereas iLQG diverges from the target.

## H    NETWORK INITIALIZATION STRATEGIES

For linear dynamics and cartpole used the Xavier initialization strategy Glorot & Bengio (2010). This is considered to be standard method for recurrent networks that use tanh activations. This strategy was crucial to allow using large learning rates and allow convergence in $\sim 2000 - 4000$ iterations. Without this strategy, convergence is extremely slow and prohibits the use for large learning rates.

On the other hand, for high dimensional systems like the quadcopter and human arm, this strategy failed to work. The reason is partially explained in the loss function section of this supplementary text. Essentially, the $FC$ layers having $\mathcal{O}(n_x^2)$ trainable parameters and random initialization values cause a snowballing effect on the propagation of $V_{\boldsymbol{x},t}$ causing the loss function to diverge and make training impossible as the gradient step in never reached. A simple solution to this problem was to use zero initialization for all weights and biases. This prevents $V_{\boldsymbol{x},t}$ from diverging as network predictions are zero and the state trajectory propagation is purely noise driven. This allows computing the loss function without diverging and taking gradient steps to start making meaningful predictions at every time step. Although this allows for training the network for high-dimensional systems, it slows down the process and convergence requires many more iterations. Therefore, further investigation into initialization strategies or other recurrent network architectures would allow improving convergence speed.

## I    MACHINE INFORMATION

We used Tensorflow Abadi et al. (2015) extensively for defining the computational graph in fig. 1 and model training. Because our current implementation involves many consecutive CPU-GPU transfers, we decided to implement the CPU version of Tensorflow as the data transfer overhead was very time consuming. The models were trained on multiple desktops computers / laptops to evaluate diffent models and hyperparameters. An Alienware laptop was used with the following specs:
Processor: Intel Core i9-8950HK CPU @ 2.90GHz×12, Memory: 32GiB.
The desktop computers used have the following specs:
Processor: Intel Xeon(R) CPU E5-1607v3 @ 3.10GHz×4 Memory: 32GiB

