# OpenReview forum: "Deep Nonlinear Stochastic Optimal Control for Systems with Multiplicative Uncertainties"
_ICLR.cc/2020/Conference — Reject_

### Official Review · AnonReviewer2 · 2019-10-13
**Official Blind Review #2**

**Rating:** 3

**Review:**

Summary and Decision

The authors in the paper studied the problem of stochastic optimal control using deep learning. The main contributions of this work can be summarized by introducing an additional backward stochastic differential equation (BSDE) over Pereira et al. (2019), where instead of predicting the gradient of the value function (V_x), the authors predicted the values required to compute the Hamiltonian H(V_x) instead. The authors then compared the results against both closed form solutions of optimal control and a numerical approximate controller by Li and Todorov (2007). However, it is unclear whether this particular modification leads to any incremental benefits over Pereira et al. (2019), nor is it compared against the feedforward networks of Han et al. (2017). Furthermore, one of the main benefits of this modifications the authors claimed is scalability to higher dimensions, yet we do not have any experiments in more than 6 dimensions in the quadcopter case.

While I believe the authors have a promising idea, the current paper do not provide enough justification to demonstrate an improvement. Therefore I recommend a weak reject.


Background

In the classical optimal control set up, it is often very difficult to recover a solution in closed form. Yet at the same time, current numerical methods are far from satisfactory. There are two main approaches to numerically solving optimal control:
1. Numerically solve the Hamilton-Jacobi-Bellman (HJB) equation for the value function, which is difficult both due to non-linearity and curse of dimensionality.
2. Use the forward-backward stochastic differential equation (FBSDE) representation of the solution for the HJB equation, but simulation of the backward equation is difficult due to requirement of meeting a terminal condition.

Han et al. (2017) worked around the simulation difficulty by instead training a feedforward network to minimize the error of terminal condition. In the same paper, Han et al. showed this method can solve an HJB equation in 100 dimensions. Pereira et al. (2019) extended this idea to recurrent networks. Bakshi et al. (2017b) introduced the second BSDE to encourage more controlled exploration, which is used in the current paper under review.

The ultimate goal of this line of work is to solve highly complex non-linear stochastic control problems, where we have no hope of recovering an optimal in closed form. Therefore, any method with efficient approximate computation is highly desirable. In particular, methods involving an approximation with deep networks are quite promising and deserve further exploration.


Detailed Comments

The authors in this paper essentially combined the idea of Bakshi et al. (2017b) and Pereira et al. (2019) in an attempt to improve both existing papers. Having a guided second order controlled BSDE will encourage further exploration, and intuitively this change should make training easier. Furthermore, by introducing the prediction of the quantity \Omega_t, the authors avoid computing a rank 3 tensor. Hence, I believe this is a promising idea and deserves to be implemented and carefully studied.

The main concern of this work is that it's not clear whether this method is working. Essentially, I would like to see that despite introducing more complexity to the network compared to Pereira et al. (2019) and Han et al. (2017), the resulting network can still be trained to achieve improved results. It is very unsatisfying to only compare against a closed form solution and a simple approximate control. These experiments can serve to show that the trained network is not behaving erratically, but we cannot conclude any improvements. In fact, even when compared against the approximate control ILQG by Li and Todorov (2007), it's unclear if the current method is better when given the same computation budget.

The other main concern is scalability to higher dimensions. While computationally cheap, it is unclear whether or not predicting the quantity \Omega_t does in fact work in high dimensions. This concern is mainly driven by the fact Han et al. (2017) has been able to solve control problems in 100 dimensions, where computing the rank 3 tensor is becoming costly. In this case, comparison against the closed form solution is sufficient to determine whether or not predicting \Omega_t works in high dimensions.


Minor Additional Comments

There are more minor points I would like to make, but these do not contribute to the review decision.
1. On page 3, below equation (1), I believe C is a map [0,T] X R^{n_x} X R^{n_u} -> R^{ n_x * (n_w + 1) }
2. On Page 3, below equation (2), it's strange seeing this notation C^{1,2} for a function of x only; in stochastic control and PDE literature, C^{1,2} typically denotes a function u(t,x) once differentiable in t and twice differentiable in x.
3. On page 3, below equation (3), we do also require the matrix R to be symmetric?
4. On page 3, for equation (4), it should be mentioned at some point in this paper that we can only recover an HJB equation without a sup term due to the loss \ell(x,u) being quadratic in the control u. In general, the HJB equation and the FBSDEs can be much more difficult to work with.
5. On page 4, it would be more satisfying to cite a more complete collection of works in stochastic control and FBSDEs. In particular, Bismut (1976) and Pardoux and Peng (1990) are seminal works that led up to El Karoui et al. (1997). It would be also nice to briefly mention a huge literature on the PDE methods to control.
6. On page 5, the notation for \psi and \zeta were not introduced. I interpreted from context that these were parameters for predicting \Gamma and \Omega, but it would nice to have a definition.

**Experience Assessment:**

I have read many papers in this area.

**Review Assessment: Checking Correctness Of Derivations And Theory:**

I assessed the sensibility of the derivations and theory.

**Review Assessment: Checking Correctness Of Experiments:**

I carefully checked the experiments.

**Review Assessment: Thoroughness In Paper Reading:**

I read the paper thoroughly.

---

> ### Author Response · Authors · 2019-11-15
> **Response to reviewer #2**
>
> We did not compare against the feedforward network because a detailed comparison was done in Pereira et. al., which demonstrated clear advantages of the LSTM-based architecture. Furthermore, for the time horizon and discretization used in our simulations (150 timesteps vs 30 in Han et. al.), having a different feedforward network at every timestep certainly won't scale in computational time and memory requirements.
>
> We'd also like to clarify that the quadcopter dynamics used in the paper contains 12 states and 4 controls, and the human arm dynamics consists of 10 states and 6 controls.
>
> When compared against prior work (FBSDE and iLQG), our proposed method achieves a much smaller variance in its trajectories for the quadcotper experiment. For the human arm system, FBSDE can not handle any control multiplicative noise. Compared against iLQG, our method demonstrates better task performance (shown in the phase plot included in the supplementary materials fig. 9), and its performance deteriorates more slowly when the noise present in the system increases (shown in the main paper fig. 5). Therefore, we conclude that the proposed controller demonstrates improved performance over prior methods inspite of added complexity to the network architecture.

---

> > ### Comment · AnonReviewer2 · 2019-11-15
> > **Additional Clarifications**
> >
> > I believe I need to clarify my points further here.
> >
> > 1. A separate experiment is needed for evaluate the effects of a guided second order BSDE for control. As this is the main modification from previous works, it's hard to judge whether seemingly minor change can lead to a significant benefit. It is entirely possible that the authors have a very novel and powerful simple idea, but at this point it's at best unclear.
> >
> > 2. The authors claimed their method can be scaled to higher dimensions because they avoid computing a rank 3 tensor (in the state variable only) and hence should be faster. While this claim is generally true when state is very high in dimension, this experiments are only in roughly 10 dimensions, and we would expect little effects from computing a rank 3 tensor. At the same time, given that Han et al. were able to perform control in 100 dimensions, I believe the authors should demonstrate their method in solving a problem of comparable size.
> >
> > 3. The main criticism of whether or not the control is better than iLQG depends on whether the two methods have the computational budget. I can always choose very crude step sizes and parameters so that the iLQG's variance is significantly larger than any method. It's important to tune the benchmark method and give it the same computational budget the LSTM has to make a valid comparison.
> >
> > To raise my score, the authors will have to address these serious questions.

---

### Official Review · AnonReviewer1 · 2019-10-19
**Official Blind Review #1**

**Rating:** 3

**Review:**


######### Rebuttal Response:
Thanks for the clarifications and especially for updating the formatting. The current state does not convince me to rate the paper as weak accept but I increased my rating to weak reject.

"Pereira et. al. has shown that a recurrent network architecture using LSTM outperforms the fully connected networks at every time step proposed by Han et. al. in task completion, space and time complexity. Therefore, in this paper we choose to use an LSTM-based network architecture."

-> Yes it might be true that a recurrent function approximator does in practice perform better than a feed-forward function approximator. However, in theory a feed-forward network should be sufficient as the value function does not depend on the previous states. Therefore, the question is, why does the LSTM perform better? Does the recurrent nature of the LSTM make the predictions smoother compared to a feed-forward network?  Can any other regularizing scheme be introduced s.t. the feed-forward networks performs equally well?

######### Review:

Summary:
The paper builds on the work of Pereira et. al. and uses forward backward stochastic differential equations to learn the Hessian of the Value function Vxx and \partial _t V_x + 1/2 tr(\partial_{xx} V_x CC^T). In contrast to the prior work, this paper introduces multiplicative noise for the control and uses second order optimization. The performance is evaluated on different control tasks, e.g., linear system, cartpole, quadcopter & human arm actuated by tendons.

Conclusion:
All in all, I like the proposed research of combining theoretical approaches and deep learning to perform trajectory optimization and I would like to see much more of this research like this within the ICLR community. Furthermore, I think that the paper has a contribution and that the paper was improved compared to the initial Neurips submission (i.e., adding ILQG as baseline). However, the writeup and formatting is still very much sub-standard and must be improved to make this paper worth publishing. The current write-up is not accessible for the ICLR community and the understandability must be significantly improved (Details are provided below). Therefore, I currently rate this paper as a clear rejection but I am happy to improve the score to 7-8 if the write up is improved during the rebuttal.


Theoretical Structure:
I like the introduction, which covers the topic but might be a bit too long. Maybe you want to shorten the introduction and add an additional related work section at the end. The stochastic control introduction is nice and has the correct level of abstraction for the reader. However, the paper introduces many complex concepts which are not essential for understanding the paper (e.g., filtered probability space etc.). One might want to trade off understandability vs. mathematical rigor especially, if the paper does not rely on these concepts. Furthermore, you might want to make eq 1 more explicit as the multiplicative action noise is not visible from eq 1. Section 3 'A FBSDE Solution to the HJB PDE' is the most problematic section of this paper, which is not understandable for the common ICLR reader. Eq. 6, which just appears without any derivation, is not understandable and the reader has no intuition how to derive this eq. Furthermore, Eq 6 (page. 4) uses notations which is only clearly introduced later within the paper or even the appendix (e.g., Y being the propagated value function, Z being the propagated gradient of the value function is only mentioned in the appendix, i.e., page 12. \Gamma is only introduced in page 5. Yes, Eq. 7 defines these variables but the style of definition is not standard and one does not expect the variables to be defined in this style.). Could the authors please provide an intuitive derivation of these equation and use clearer notation (Why would one want to abstract V, V_x, V_x, \mathcal|{H}(V_x) in the first place as these are intuitive for the ICLR community and sufficiently short?) Especially, as this section highlights the difference to the prior works of Pereira et. al., this section should be very clear. Section 4 is clear but should include the loss function as the loss is not trivial and essential for the optimization. Currently, the loss description is buried in the appendix. All in all, the theoretical explanation and the bloated notation should be simplified and every equation should be embedded into an intuitive derivation. Currently these explanations are not understandable without reading the appendix and prior work.


Experiments:
The experiments apply 2FBSDE to 4 different control tasks (Linear system, quadcopter, cartpole & human arm) and compare the performance to the prior work of FBSDE and iLQG. The number of baselines and systems is sufficient. However, the paper should provide more evaluations:

(1) Plot the histogram of the obtained cost distributions.
(2) Plot a single state- and action trajectory (and the action distribution). Using these plots, the level of noise and smoothness and hence the applicability to physical systems can be evaluated.
(3) Plot the noise free trajectories and show that these mean trajectories reach the desired solution.
(4) Specify the exact cost function for every experiment

Further Comments to the individual experiments:

Cartpole:
The Cartpole iLQG seems to perform much better (swing-up the pendulum faster, don't deviate so much from x=0, much more coherent velocity compared to 2FBSDE, FBSDE). Could the authors please discuss these aspects in more detail and present experiments with longer time-horizons to check whether the proposed method can stabilize the cart at [0, 0, 0, 0]. The current plots don't reach this target state. Your plots also hint that the cartpole does not need to pre-swing the pendulum, which is most likely due to the very low action cost. This selection of action cost significantly simplifies the problem. Could the authors please include a cartpole with higher action cost and show that 2FBSDE can learn to pre-swing the pendulum.

The quadcopter:
Could the authors please specify the exact quadcopter dynamics. What kind of abstraction did you model? What are the control inputs? Furthermore, the citation for the dynamics is wrong and puts the supervisor of the master thesis as first author. Furthermore, can the authors please provide longer plots to highlight, which method can stabilize the system.

Human Arm:
For the human arm neither iLQG or 2FBSDE reach the desired target location. Can you explain why no trajectory optimization does reach the desired position.

Formatting:
Please rework the formatting such that the inline math does not cause the formatting issues of different line spacings (e.g., sec. 2.1, sec. 5) and irregular whitespaces (e.g., last line of paragraph 2.1 Preliminaries). Please remove the color coding of text for the experiments and make sure that the legends are sufficiently large and include all lines. Currently the legends are missing the target state. You can also extend the figure captions to highlight the conclusion of the plots. Please rework the figures such that the figures do not cause so much whitespace (e.g., Figure 3, 4 & 5). When reconfiguring the plots, you gain space, which can be used for further explanation of the theory. Furthermore, you might want add dotted lines to the confidence intervals as the confidence intervals are important but the differences are not clearly visible from the plots. Also, the labelling in figure 3 seems wrong, the axis labeled cart velocity should be pendulum angle and the pendulum angle axis should be cart velocity. All in all, the formatting can be significantly improved, which is especially bad as this paper is most likely a resubmission from Neurips.


Minor Comments / Questions:
- 'where l :Rnx×Rnu→R+ is the running cost and C1,23 \phi :Rnx→R+ is the terminal state cost.' Is l and \phi of class C^{1,2} or only one of them? The notation is confusing and should be simplified.
- Can you comment on how important this multiplicative noise in physical system? ¬¬¬
- Why are you using a LSTM instead of a simple feed-forward neural network as the ff-nn should be sufficient to model V(x) as the value function is not recurrent. Have you tried using a simple ff-nn?

**Experience Assessment:**

I have published one or two papers in this area.

**Review Assessment: Checking Correctness Of Derivations And Theory:**

I assessed the sensibility of the derivations and theory.

**Review Assessment: Checking Correctness Of Experiments:**

I carefully checked the experiments.

**Review Assessment: Thoroughness In Paper Reading:**

I read the paper thoroughly.

---

> ### Author Response · Authors · 2019-11-15
> **Response to reviewer #1**
>
> The control multiplicative noise can be found in bio-mechanical systems [1] and financial problems such as portfolio optimization [2].
>
> Pereira et. al. has shown that a recurrent network architecture using LSTM outperforms the fully connected networks at every time step proposed by Han et. al. in task completion, space and time complexity. Therefore, in this paper we choose to use an LSTM-based network architecture.
>
> After performing the comparison of 2FBSDE and iLQG under different levels of additive and multiplicative noise (results shown in fig. 5), we included the phase plot comparison of the two controllers in the low noise condition (multiplicative noise std = additive noise std = 0.1) in the supplementary materials (fig. 9). Under this condition, 2FBSDE can perform the task while iLQG diverges. As the noise level increases, the performance of 2FBSDE deteriorates but still outperforms iLQG.
>
> Our cost functions for all tasks were quadratic state (both running and terminal) and control costs. We have added the values of state cost and control cost coefficients for each simulation experiment to the supplementary materials. All the tasks were reaching tasks meaning that the state costs are squared distances from a target state. We refer the reviewer to sections E and F of the supplementary materials for the exact values of the parameters used in our experiments.
>
> Regarding suggestions for additional plots and experiments, we have included new plots in the main paper and supplementary materials (see section G).
>
> [1] Todorov, Emanuel. "Stochastic optimal control and estimation methods adapted to the noise characteristics of the sensorimotor system." Neural computation 17.5 (2005): 1084-1108.
>
> [2] Davis, Mark, and Sebastien Lleo. "Jump-diffusion risk-sensitive asset management II: jump-diffusion factor model." SIAM Journal on Control and Optimization 51.2 (2013): 1441-1480.

---

### Official Review · AnonReviewer3 · 2019-10-23
**Official Blind Review #3**

**Rating:** 6

**Review:**

**Summary:** The paper contains a method tailored to a certain kind of SOC problems involving multiplicative noise. The central idea is to use a recurrent network to transform the observations into a representation that can be used with solvers specifically tailored towards that class of problems.

**Decision:** I recommend to accept the paper for publication.

**Arguments for decision:** The paper clearly adresses an important problem and poroposes a method capable of solving it. The method appears to be theoretically founded and the experimental validation seems solid. The relevance of the method is there as the problem class is prevalent in practical applications. The venue is a good fit as well, as the focus is the representation of a control problem in a way that allows more efficient solutions.

**Feedback for improvement:**

- The type setting could be improved at times. E.g. below equation (1).
- I feel that the term "exploration" is overloaded. While it serves as an explicit mean to reduced the sample complexity of methods in RL, it appears to be about avoiding premature convergence in this work. I am too unfamiliar with the relevant SOC literature to judge how well the term fits, but coming from a ML background I stumbled over this expression.
- Some of the experimental details, e.g. the exact choice of time discretisation, don't appear motivated well.
- The paper needs to respect [1, 2] in the related work and show relations. From the perspective of learning state representations for optimal control, both works are relevant.
- Is it necessary to start the discussion from the continuous case? While I appreciate the elegance of starting out with a continuous problem and then discretising at the last step, it felt like a barrier to understanding in my case, as my understanding of continuous optimal control is limited–and I feel the audience of ICLR might have the same problem.

**References:**

[1] Watter, Manuel, et al. "Embed to control: A locally linear latent dynamics model for control from raw images." *Advances in neural information processing systems*. 2015.
[2] Banijamali, Ershad, et al. "Robust locally-linear controllable embedding." *arXiv preprint arXiv:1710.05373*(2017).

**Experience Assessment:**

I have read many papers in this area.

**Review Assessment: Checking Correctness Of Derivations And Theory:**

I assessed the sensibility of the derivations and theory.

**Review Assessment: Checking Correctness Of Experiments:**

I assessed the sensibility of the experiments.

**Review Assessment: Thoroughness In Paper Reading:**

I read the paper at least twice and used my best judgement in assessing the paper.

---

> ### Author Response · Authors · 2019-11-15
> **Reponse to reviewer #3**
>
> We want to clarify that the main contribution of the paper is the introduction of a recurrent neural network architecture tailored to solve a novel representation (second order FBSDEs with control) of the stochastic optimal control problem involving nonlinear systems wherein noise entering the system has a multiplicative effect with the applied controls. We work with known dynamics and full state information, hence the papers [1, 2] mentioned by the reviewer are not relevant to the problem we propose. Our 2FBSDE controller does not extract or rely on a latent representation of the observations.
>
> Regarding the exact choice of time discretization, the values were hand-tuned until we observed numerical stability, convergence of training (or optimization in case of iLQG) and reasonable task performance. We would like to reiterate that in the case of iLQG, finer time discretizations were required as compared to FBSDE and 2FBSDE. As far as the linear system time discretization, we used the same value as in Bakshi et al. (2017).
>
> Since our framework relies on the transformation of the HJB PDE to 2FBSDEs which in turn relies on mathematical results from stochastic calculus of continuous time systems, it is therefore necessary to start with the continuous time representation and only discretize the problem at the very end.

---

> > ### Comment · AnonReviewer3 · 2019-11-15
> > **Disagreement on the relevance of the related work**
> >
> > Just to make one thing clear: I am not affiliated with neither those works nor their authors.
> >
> > But your work turns a problem (specified by known system equations) into a form tailored towards a certain class of solvers. E2C and RCE do the same, except that the problem is not specified by system equations but through data.
> >
> > The claim that the work is not related is just wrong, especially given the community that you want to adress. If your work is not related to this method (the major difference being learning the necessary representation from data, which you don't do) I wonder why you send your article to a conference on learning representations anyway.

---

> > > ### Author Response · Authors · 2019-11-15
> > > **Clarifying our initial comment on relevance of related work**
> > >
> > > We understand the reviewer's opinion on this matter. There are different perspectives in terms of what prior work is relevant and what is not. One perspective is related to the general theme of approximate dynamic programming and model-based RL. There exist a plethora of prior work on approximate dynamic programming that is based on function approximations method for model learning such LWPR, Gaussian Processes, Mixture Models and trajectory optimization/optimal control, including the citations proposed by the reviewer.
> > >
> > > Our perspective however on the relevance with respect to prior work is more specific and relies on the methodological characteristics and similarities of our approach with others. We feel that in order to better facilitate audience’s understanding of our approach, we choose to familiarize our readers with the recent works that share a similar methodology (FBSDE. 2FBSDE, deep FBSDE, etc.) for stochastic optimal control. This helps us maintain a coherent introduction to this area of research and smoothly transition to the contributions of this paper.

---

### Decision · Program_Chairs · 2019-12-19

**Decision:**

Reject

**Comment:**

A nice paper, but quite some unclarities; it's unclear  in particular if the paper improves w.r.t. SOTA.  Esp. scaling is an issue here. Also, the understandability is below par and more work can make this into an acceptable submission.